# Reduced global warming from CMIP6 projections when weighting models by performance and independence

Lukas Brunner[1], Angeline G. Pendergrass[2,1*], Flavio Lehner[1*], Anna L. Merrifield[1], Ruth Lorenz[1], and Reto Knutti[1]

[1]Institute for Atmospheric and Climate Science, ETH Zurich, Zurich, Switzerland
[2]National Center for Atmospheric Research, Boulder, CO, USA
[*]Now at: Department of Earth and Atmospheric Sciences, Cornell University, Ithaca, NY, USA
**Correspondence:** Lukas Brunner (lukas.brunner@env.ethz.ch)

**Abstract.** The sixth Coupled Model Intercomparison Project (CMIP6) constitutes the latest update on expected future climate change based on a new generation of climate models. To extract reliable estimates of future warming and related uncertainties from these models, the spread in their projections is often translated into probabilistic estimates such as mean and likely range. Here, we use a model weighting approach, which accounts for the models' historical performance based on several

diagnostics as well as model inter-dependence within the CMIP6 ensemble, to calculate constrained distributions of global mean temperature change. We investigate the skill of our approach in a perfect model test, where we use previous-generation CMIP5 models as pseudo-observations in the historical period. The performance of the so weighted distribution in matching the pseudo-observations in the future is then evaluated and we find a mean increase in skill of about $17\%$ compared to the unweighted distribution. In addition, we show that our independence metric correctly clusters models known to be similar

based on a CMIP6 "family tree", which enables applying a weighting based on the degree of inter-model dependence. We then apply the weighting approach, based on two observational estimates (ERA5 and MERRA2), to constrain CMIP6 projections in weak (SSP1-2.6) and strong (SSP5-8.5) climate change scenarios. Our results show a reduction in projected mean warming for both scenarios because some CMIP6 models with high future warming receive systematically lower performance weights. The mean of end-of-century warming (2081-2100 relative to 1995-2014) for SSP5-8.5 with weighting is $3.7\,°C$, compared to

$4.1\,°C$ without weighting; the likely ($66\%$) uncertainty range is $3.1\,°C$ to $4.6\,°C$, a decrease in spread of $13\%$. For SSP1-2.6, weighted end-of-century warming is $1\,°C$ ($0.7\,°C$ to $1.4\,°C$) a reduction of $-0.2\,°C$ in the mean and $-24\%$ in the likely range compared to the unweighted case.

## 1  Introduction

Projections of future climate by Earth System Models provide a crucial source of information for adaptation planing, mitigation

decisions, and the scientific community alike. Many of these climate model projections are coordinated and provided within the frame of the Coupled Model Intercomparison Projects (CMIPs), which are now in phase 6 (Eyring et al., 2016). A typical way of communicating information from such multi-model ensembles (MMEs) is through a best estimate and an uncertainty range or a probabilistic distribution. In doing so it is important to make sure that the different sources of uncertainty are identified,

discussed, and accounted for, to provide reliable information without being overconfident. In climate science typically three main sources of uncertainty are identified in MMEs: (i) uncertainty in future emissions, (ii) internal variability of the climate system, and (iii) model response uncertainty (e.g., Hawkins and Sutton, 2009; Knutti et al., 2010).

Uncertainty due to future emissions can easily be isolated by making projections conditional on scenarios such as the Shared Socioeconomic Pathways (SSPs) in CMIP6 (O'Neill et al., 2014) or the Representative Concentration Pathways (RCPs) in CMIP5 (van Vuuren et al., 2011). The other two sources of uncertainty are harder to quantify since reliably separating them is often challenging (e.g., Kay et al., 2015; Maher et al., 2019). Model uncertainty (sometimes also referred to as structural uncertainty or response uncertainty) is used here to describe the differing responses of climate models to a given forcing due to their structural differences following the definition by Hawkins and Sutton (2009). Such different responses to the same forcing can emerge, among other things, due to different processes and feedbacks as well as due to the parametrisations used in the different models (e.g., Zelinka et al., 2020). Internal variability, finally, here refers to a model's sensitivity to the initial conditions as captured by initial-condition ensemble members (e.g., Deser et al., 2012). In this sense, it stems from the chaotic behavior of the climate system at different time scales and is highly dependent on the variable of interest as well as the period and region considered. While, for example, uncertainty in global mean temperature is mainly dominated by differences between models, regional temperature trends are considerably more dependent on internal variability. Recently, efforts have been made to use so-called Single Model Initial-condition Large Ensembles (SMILEs) to investigate internal variability in the climate projections more comprehensively (e.g., Kay et al., 2015; Maher et al., 2019; Lehner et al., 2020; Merrifield et al., 2020).

Depending on the composition of the investigated MME, uncertainty estimates often fail to reflect that included models are not independent from each other. In the development process of climate models, ideas, code and even full components are shared between institutions or models might be branched from each other in order to investigate specific questions. This can lead to some models (or model components) being copied more often, resulting in an over-representation of their respective internal variability or sensitivity to forcing (Masson and Knutti, 2011; Bishop and Abramowitz, 2013; Knutti et al., 2013; Boé, 2018; Boé and Terray, 2015). The CMIP MMEs in particular have not been designed with the aim of including only independent models and are therefore sometimes referred to as "ensembles of opportunity" (e.g., Tebaldi and Knutti, 2007) incorporating as many models as possible. When calculating probabilities based on such MMEs it is therefore important to account for model inter-dependence in order to accurately translate model spread into estimates of mean change and related uncertainties (Knutti, 2010; Knutti et al., 2010).

In addition, not all models represent the aspects of the climate system relevant to a given question equally well. To account for that, a variety of different approaches have been used to weight, sub-select, or constrain models based on their historical performance. This has been done both regionally and globally as well as for a range of different target metrics such as end-of-century temperature change or Transient Climate Response (TCR) (for an overview see, e.g., Knutti et al., 2017a; Eyring et al., 2019; Brunner et al., 2020b). Global mean temperature increase in particular is one of the most widely discussed effects of continuing climate change and the main focus of many public and political discussions. With the release of the new generation of CMIP6 models, this discussion has been sparked yet again, as several CMIP6 models show stronger warming than most of the earlier-generation CMIP5 models (Andrews et al., 2019; Gettelman et al., 2019; Golaz et al., 2019; Voldoire et al.,

2019; Swart et al., 2019; Zelinka et al., 2020; Forster et al., 2020). This raises the question of whether these models are accurate representations of the climate system and what that means for the interpretation of the historical climate record and the expected change due to future anthropogenic emissions.

Here, we use the Climate model Weighting by Independence and Performance (ClimWIP) method (e.g., Knutti et al., 2017b; Lorenz et al., 2018; Brunner et al., 2019; Merrifield et al., 2020) to weight models in the CMIP6 MME. Weights are based on (i) each model's performance in simulating historical properties of the climate system such as horizontally resolved anomaly, variability, and trend fields, and (ii) its independence from the other models in the ensemble, estimated based on shared biases of climatology. In contrast to many other methods, which constrain model projections based on only one observable quantity, such as the warming trend (e.g., Giorgi and Mearns, 2002; Ribes et al., 2017; Jiménez-de-la Cuesta and Mauritsen, 2019; Liang et al., 2020; Nijsse et al., 2020; Tokarska et al., 2020), ClimWIP is based on multiple diagnostics, representing different aspects of the climate system. These diagnostics are chosen to evaluate a model's performance in simulating observed climatology, variability, and trend patterns. Note that, in contrast to other approaches such as emergent constraint-based methods, some of these diagnostics might not be highly correlated with the target metric (however, it is still important that they are physically relevant – to avoid introducing noise without useful information in the weighting). Combining a range of relevant diagnostics is less prone to overconfidence, since the risk of up-weighting a model because it "accidentally" fits observations for one diagnostic, while being far away from them in several others is greatly reduced. In turn, methods which are based on such a basket of diagnostics have been found to generally lead to weaker constraints (Sanderson et al., 2017; Brunner et al., 2020b), as the effect of the weighting typically weakens when adding more diagnostics (Lorenz et al., 2018).

ClimWIP has already been used to create estimates of regional change and related uncertainties for a range of different variables such as Arctic sea ice (Knutti et al., 2017b), Antarctic ozone concentrations (Amos et al., 2020), North American maximum temperature (Lorenz et al., 2018) and European temperature and precipitation (Brunner et al., 2019; Merrifield et al., 2020). Recently, Liang et al. (2020) have used an adaptation of the method to constrain changes in global temperature using global mean temperature trend as single diagnostic for both the performance and independence weighting. Here, we focus on investigating the ClimWIP method's performance in weighting global mean temperature changes when informed by a range of diagnostics. To assess the robustness of these choices, we perform an out-of-sample perfect model test using CMIP5 and CMIP6 as pseudo-observations. Based on these results, we select a combination of diagnostics which capture not only a model's transient warming but also its ability to reproduce historical patterns in climatology and variability fields in order to increase the robustness of the weighting scheme and minimize the risk of skill decreases due to the weighting. This approach is particularly important for users interested in the "worst case" rather than in mean changes. We also look into the inter-dependencies among the models, showing the ability of our diagnostics in clustering models with known shared components using a "family tree" (Masson and Knutti, 2011; Knutti et al., 2013) and further the skill of the independence weighting to account for this. We then calculate combined performance-independence weights based on two reanalysis products in order to also account for the uncertainty in the observational record. Finally, we apply these weights to provide constrained distributions of future warming and TCR.

## 2 Data and Methods

### 2.1 Model data

The analysis is based on all currently available CMIP6 models which provide surface air temperature (tas) and sea level pressure (psl) for the historical, SSP1-2.6, and SSP5-8.5 experiments. We use all available ensemble members, which is a total of 129 runs from 33 models (see table S4 in the supplementary material for a full list including references). We use models post-processed within the ETH Zurich CMIP6 next generation archive, which provides additional quality checks and re-grids models onto a common $2.5° \times 2.5°$ latitude-longitude grid, using second order conservative remapping (see Brunner et al., 2020a, for details). In addition, we use one member of all CMIP5 models providing the same variables and the corresponding experiments (historical, RCP2.6, RCP8.5) which is a total of 27 models (see table S5 for a full list).

### 2.2 Reanalysis data

To represent historical observations in tas and psl, we use two reanalysis products: ERA5 (C3S, 2017) and MERRA2 (GMAO, 2015a, b; Gelaro et al., 2017). Both products are regridded to a $2.5° \times 2.5°$ latitude-longitude grid using second order conservative remapping and are evaluated in the period 1980-2014. We use a combination of these two observational datasets following the results of Lorenz et al. (2018) and Brunner et al. (2019), who show that using individual datasets separately can lead to diverging results in some cases. It has been argued that that combining multiple datasets (e.g., by using their full range or their mean) yields more stable results (Gleckler et al., 2008; Brunner et al., 2019). Here we use the mean of ERA5 and MERRA2 at each grid point as reference equivalent to Brunner et al. (2019). Finally, we also compare our results to globally averaged merged temperatures from the Berkley Earth Surface Temperature (BEST) data set (Cowtan, 2019).

### 2.3 Model weighting scheme

We use an updated version of the ClimWIP method described in Brunner et al. (2019) and Merrifield et al. (2020), which is based on earlier work by Lorenz et al. (2018), Knutti et al. (2017b), Sanderson et al. (2015b), and Sanderson et al. (2015a); it can be downloaded at: https://github.com/lukasbrunner/ClimWIP.git. It assigns a weight $w_i$ to each model $m_i$ that accounts for both model performance as well as independence,

$$w_i = \frac{e^{-\left(\frac{D_i}{\sigma_D}\right)^2}}{1 + \sum_{j \neq i}^{M} e^{-\left(\frac{S_{ij}}{\sigma_S}\right)^2}}, \tag{1}$$

where $D_i$ and $S_{ij}$ are the generalised distances of model $m_i$ to the observations and to model $m_j$, respectively. The shape parameters $\sigma_D$ and $\sigma_S$ set the strength of the weighting, effectively determining the point at which a model is considered to be "close" to the observations or to another model (c.f., section 2.5).

This updated version of ClimWIP assigns the same weight to each initial-condition ensemble member of a model, which is adjusted by the number of ensemble members (see Merrifield et al., 2020, for a detailed discussion). To illustrate this additional

step in the weighting method, consider a single performance diagnostic $d$. $d$ is calculated for each model and ensemble member separately, hence $d = d_i^k$ with $i$ representing individual models and $k$ running over all ensemble members $K_i$ of model $m_i$ (from one to 50 members in CMIP6). For each model $m_i$, the mean diagnostic $d_i'$ is,

$$d_i' = \frac{\sum_k^K d_i^k}{K_i} \tag{2}$$

$d_i'$ is then used to calculate the generalised distance $D_i$ and further the performance weight $w_i$ via (1). A detailed description of this processing chain can be found in section S2 in the supplement. An analogous process is used for distances between models. This setup allows a consistent comparison of model fields to each other and to observations in the presence of internal variability and, in particular, also enables the use of variance-based diagnostics. In addition, it ensures a consistent estimate of the performance shape parameter $\sigma_D$ in the calibration (see section 2.5), based on the average weight per model; in previous work, in contrast, the calibration was based on only one ensemble member per model.

## 2.4  Weighting target and diagnostics

We apply the weighting to projections of annual mean, global mean temperature change from two SSPs, representing weak (SSP1-2.6) and strong (SSP5-8.5) climate change scenarios. Changes in two 20-year target periods representing mid-century (2041-2060) and end-of-century (2081-2100) conditions are compared to a 1995-2014 baseline. In addition, we weight TCR values obtained from an update of the data set described in Tokarska et al. (2020). The weights are calculated from global, horizontally-resolved diagnostics based on annual mean data in the 35-year period 1980-2014. We use different diagnostics for the calculation of the independence and performance parts of the weighting, as proposed in Merrifield et al. (2020).

The goal of the independence weighting is to identify structural similarities between models (such as shared offsets or similar spatial patterns) which are interpreted to be indications of inter-dependence arising from, e.g., shared components or parametrisations. In the past, combinations of horizontally-resolved regional temperature, precipitation, and sea level pressure fields, have typically been used (e.g., Knutti et al., 2013; Sanderson et al., 2017; Boé, 2018; Lorenz et al., 2018; Brunner et al., 2019). Building on the work of Merrifield et al. (2020), we use a combination of two global, climatology-based diagnostics, the spatial pattern of climatological temperature (tasCLIM) and sea level pressure (pslCLIM), as similar diagnostics were found to work well for clustering CMIP5-generation models known to be similar. Beside our approach, several other methods to tackle this issue of model dependence exist. Among them are approaches which use other metrics to establish model independence (e.g., Pennell and Reichler, 2011; Bishop and Abramowitz, 2013; Boé, 2018), which select a more independent sub-set of the original ensemble (e.g., Leduc et al., 2016; Herger et al., 2018a), or even treat model similarity as an indication for robustness and give models which are closer to the multi model mean more weight (e.g., Giorgi and Mearns, 2002; Tegegne et al., 2019). Neither of these definitions of independence hold in a strictly statistical sense (Annan and Hargreaves, 2017), but we still stress that it is important to account for different degrees of model inter-dependencies as good as possible when developing probabilistic estimates from an "ensemble of opportunity" such as CMIP6. Additional discussion about our method to calculate model independence in the context of other approaches can be found in section S4 of the supplement.

The performance weighting, in turn, allocates more weight to models which better represent the observed behavior of the
climate system as measured by the diagnostics, while down-weighting models with large discrepancies from the observations.
We use multiple diagnostics to limit overconfidence in the case where a model fits the observations well in one diagnostic by
chance, while being far away from them in several others. For example, we want to avoid giving heavy weight to a model
based solely on its representation of the temperature trend if its year-to-year variability differs strongly from observed year-
to-year variability. The performance weights are based on five global, horizontally-resolved diagnostics: temperature anomaly
(tasANOM; calculated from tasCLIM by removing the global mean), temperature variability (tasSTD), pslANOM, and pslSTD
as well as temperature trend (tasTREND). A detailed description of the diagnostic calculation can be found in section S2 in
the supplement. We use anomalies instead of climatologies in the performance weight in order to avoid punishing models
for absolute bias in global-mean temperature and pressure, because these are not correlated with projected warming (Flato
et al., 2013; Giorgi and Coppola, 2010). This can be different for regional cases, where, e.g., absolute temperature biases have
been shown to be important for constraining projections of Arctic sea ice extent (Knutti et al., 2017b) or European summer
temperatures (Selten et al., 2020).

One aim of our study is to find an optimal combination of diagnostics that successfully constrains projections for our target
quantity (global temperature change) while avoiding overconfidence or susceptibility to uncertainty from internal variability.
For example, tasTREND is a powerful diagnostic because of its clear physical relationship to and high correlation with pro-
jected warming (e.g., Nijsse et al., 2020; Tokarska et al., 2020). However, while it has the highest correlation to the target
of all investigated diagnostics, it also has the largest uncertainty due to internal variability (i.e., spread of tasTREND across
ensemble members of the same model). Ideally, a performance weight is reflective of underlying model properties and does not
depend on which ensemble member is chosen to represent that model. tasTREND does not fulfil this requirement: the spread
within one model is the same order of magnitude as the spread among different models. To find a compromise, we divide
our diagnostics into two groups: trend-based diagnostics (tasTREND) and not-trend based diagnostics (tasANOM, tasSTD,
pslANOM, and pslSTD). Different combinations of these two groups (ranging from only not-trend based to only tasTREND)
are evaluated in section 3.1 and the best performing combination is selected for the remainder of the study.

## 2.5 Estimation of the shape parameters

The shape parameters $\sigma_D$ and $\sigma_S$ are two constants which determine the width of the Gaussian weighting functions for all
models. As such they are responsible for translating the generalised distances into weights. In case of the performance weight-
ing, small values of $\sigma_D$ lead to aggressive weighting with a few models receiving all the weight, while large values lead to more
equal weighting. It is important to note that, while $\sigma_D$ sets this "strength" of the weighting, the rank of a model (i.e., where it
lies on the scale from best to worst) is purely based on its generalised distance to the observations. To estimate a performance
shape parameter $\sigma_D$ that weights models based on their historical performance without being overconfident, we use a calibra-
tion approach based on the perfect model test in Knutti et al. (2017b) and detailed in section S3 in the supplement. In short,
the calibration selects the smallest $\sigma_D$ value (hence the strongest weighting) for which $80\%$ of "perfect models" fall within
the 10-90 percentile range of the weighted distribution in the target period. Smaller $\sigma_D$ values lead to less models fulfilling

this criterion and hence to too narrow, overconfident projections. Note that methods that simply maximize correlation of the weighted mean to the target often tend to pick small values of $\sigma_D$ that result in projections that are overconfident in the sense that the uncertainty ranges are too small (Knutti et al., 2017b). A similar issue arises for methods which estimate $\sigma_D$ based only on historical information as better performance in the base state does not necessarily lead to a more skill representation of the future, e.g., if the chosen diagnostics are not relevant for the target (Sanderson and Wehner, 2017).

The independence weighting has a subtle but fundamentally different dependence on its shape parameter $\sigma_S$: small values lead to equal weighting, as all models are considered to be independent, but so do large values, as all models are considered to be *de*pendent. Hence, the effect of the independence weighting is strongest if the shape parameter is chosen such that it identifies clusters of models as similar (down-weighting them) while still correctly identifying models which are far from each other as independent (hence giving them relatively more weight). For a detailed discussion including SMILEs see Merrifield et al. (2020). To estimate $\sigma_S$, we use the information from models with more than one ensemble member. Simply put, we know that initial-condition ensemble members are copies of the same model that differ only due to internal variability, and therefore we have some information about the distances that must be considered "close" by $\sigma_S$. The method for calculating $\sigma_S$ is described in detail in section 3 of the supplement of Brunner et al. (2019). Here, we arrive at a value of $\sigma_S = 0.54$, which we use throughout the manuscript. It is worth noting that $\sigma_S$ is based only on historical model information, and is therefore independent from observations or the selected target period and scenario. Additional discussion of the selected $\sigma_S$ value in the context of the multi-model ensemble used in this study can be found in the supplement (section S5).

## 2.6 Validation of the performance weighting

To investigate the skill of ClimWIP in weighting CMIP6 global mean temperature change and the effect of the different diagnostic combinations, we apply a perfect model test (Abramowitz and Bishop, 2015; Boé and Terray, 2015; Sanderson et al., 2017; Knutti et al., 2017b; Herger et al., 2018a, b; Abramowitz et al., 2019). As a skill measure, we use the continuous ranked probability skill score (CRPSS), a measure for ensemble forecast quality, defined as the relative error between the distribution of weighted models and a reference (Hersbach, 2000). Here, we use the relative CRPSS change between the unweighted and weighted cases (in %), with positive values indicating a skill increase. The CRPSS is calculated separately for both SSPs and future time periods, since we expect to find different skill for different projected climate states.

The first perfect model test only focuses on the relative skill differences when applying performance weights based on different combinations of diagnostics (results are presented in section 3.1). We explain its implementation based on an example perfect model $m_j$ with only one ensemble member for simplicity here: (i) the model $m_j$ is taken as pseudo-observation and removed from the CMIP6 MME; (ii) the output from $m_j$ during the historical diagnostic period (1980-2014) is used to calculate the performance diagnostics for the remaining models ($d'_{i \neq j}$); (iii) the generalised model-"observation" distances ($D_{i \neq j}$) and the performance weights ($w_{i \neq j}$) are calculated and applied to the MME (excluding $m_j$); (iv) the CRPSS is calculated in the target periods using the future projections of $m_j$ as reference. This is done iteratively, using each model in CMIP6 MME in turn as pseudo-observation. For perfect models with more than one ensemble member ($m_j^k$), all members are removed from

the ensemble in (i), $d'_{i \neq j}$ is calculated for each member separately in (ii) and then averaged, and the CRPSS is also calculated for each ensemble member in (iv) and averaged.

This approach is structurally similar to the one used to calibrate the performance shape parameter $\sigma_D$ as integral part of ClimWIP (described in section 2.5). However, the metric and aim of this perfect model test are quite different. It is used to show the potential for a skill increase through the performance weighting, as well as the risk of a decrease based on the selected $\sigma_D$ and to establish the most skilful combination of diagnostics.

The second perfect model test (section 3.2) is conceptually similar, but pseudo-observations are now drawn from CMIP5 instead of CMIP6. This test has the advantages that the perfect models have not been used to estimate $\sigma_D$ and can be considered independent. Even though one might argue that also the CMIP5 pseudo-observations are not fully out-of-sample as several CMIP6 models are related to CMIP5 models and might be structurally similar to their predecessors, which was the case for the CMIP5 and 3 generations (Knutti et al., 2013). However, there are also considerable differences between CMIP5 and 6 that arise from many years of additional model development, a longer observational record to calibrate to, and differing spatial resolutions. In addition, the emission scenarios that force CMIP5 and 6 in the future (RCPs and SSPs, respectively) result in slightly different radiative forcings (Forster et al., 2020) and several CMIP6 have been shown to lead to considerably more warming than most CMIP5 models. We do not discuss these similarities and differences between the model generations in detail here; instead we simply use CMIP5 as a source for pseudo-observations to evaluate the skill of ClimWIP in weighting the CMIP6 MME. To avoid cases with the highest potential of remaining dependence between generations we exclude CMIP6 models which are direct predecessors of the respective CMIP5 model used as pseudo observations (see table S5 for a list).

## 2.7 Validation of the independence weighting

To validate that the information in the diagnostics chosen for the independence weighting (tasCLIM and pslCLIM) can identify models known to be similar, we use a hierarchical clustering approach based on Müllner (2011) and implemented in the Python SciPy package (www.scipy.org). We use the linkage function with the average method applied to the horizontally-resolved distance fields between each pair of models (see section S6 in the supplement for more details). This approach is conceptually similar to the work from Masson and Knutti (2011) and Knutti et al. (2013) and follows their example of showing similarity as model "family trees". The hierarchical clustering is *not* used in the model weighting itself; we use it here only to show that qualitative information about model similarity can be inferred from model output using the two chosen diagnostics and to compare it to the results from the independence weighting.

The independence weighting (denominator in equation (1)) quantifies the similarity information extracted from the pairwise distance fields via the independence shape parameter ($\sigma_S$; see section 2.5). The independence weighting estimates where two models fall on the spectrum from completely independent to completely redundant and weights them accordingly. In order to test this approach, we successively add artificial "new" models into the CMIP6 MME: for an example model with two members ($m_j^1$ and $m_j^2$), we remove the first member and add it as additional model ($m_{M+1}$). In an idealized case, where all models are perfectly independent from each other and all ensemble members of a model are identical, we would expect the weight of the member that remains ($m_j^2$) to go down by a factor 1/2, while the weight of all other models would stay the same. However, in

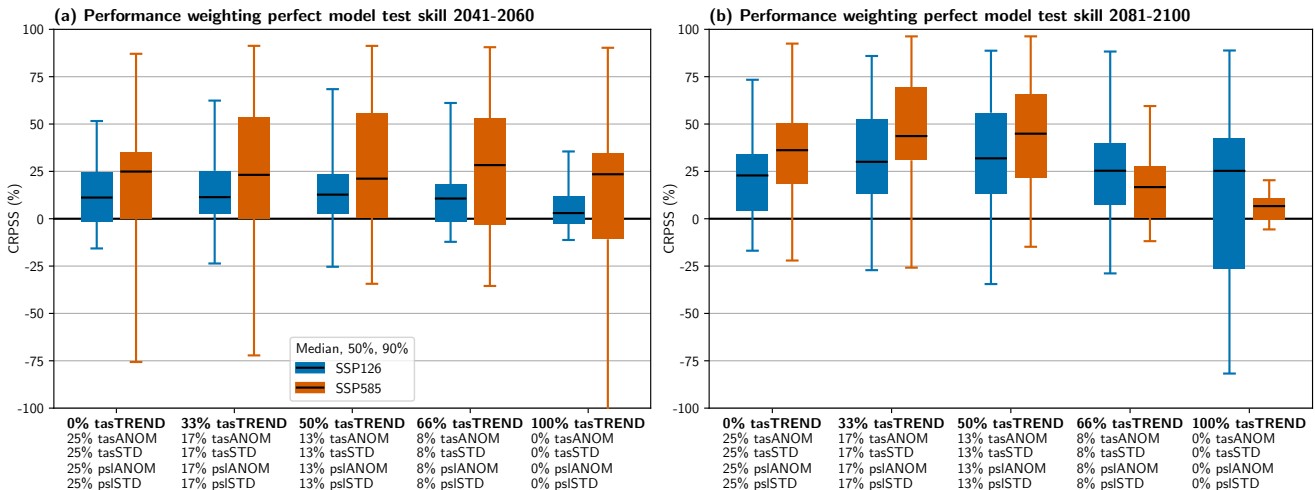

**Figure 1.** Continuous ranked probability skill score (CRPSS) relative to the unweighted ensemble for the performance weighting based on a leave-one-out perfect model test with CMIP6 for (a) mid-century and (b) end-of-century temperature change relative to 1995-2014. The x-axis shows different combinations of the two diagnostic groups ranging from only not-trend based (0 % tasTREND) to only trend-based (100 % tasTREND). Values not summing to 100 % is due to rounding in the labels only.

a real MME, where there is internal variability and complex model inter-dependencies exist, we would not necessarily expect such simple behaviour; several other models might also be (rightfully) affected by adding such a duplicate while the effect on the $m_j^2$ would be smaller (see section 4.2)

## 3    Evaluation of the weighting in the perfect model test

### 3.1    Leave-one-out perfect model test with CMIP6

We start by calculating the performance weights in the diagnostic period (1980-2014) in a pure model world and without using the independence weighting. In this first step we focus on relative skill differences when using different combinations of diagnostics. Figure 1 shows the distribution of the CRPSS (with positive values indicating an increase in projection skill due to the weighting and vice versa; see section 2.6) evaluated for two the mid- and end-of-century target periods, the two SSPs, and for different combinations of diagnostics. The diagnostics range from only not-trend based (0 % tasTREND + 25 %

tasANOM + 25 % tasSTD + 25 % pslANOM + 25 % pslSTD = 100 %) to only tasTREND based (100 % tasTREND). Overall, all diagnostic combinations tend to increase median skill compared to the unweighted projections, but there is a considerable range of CRPSS values and they can be negative. In evaluating the different cases we hence focus on two important aspects of the CRPSS distribution: (i) the median as best estimate of expected relative skill change and (ii) the 5th and 25th percentiles in particular if they are negative. Negative CRPSS values indicate a worsening of the projections compared to the unweighted

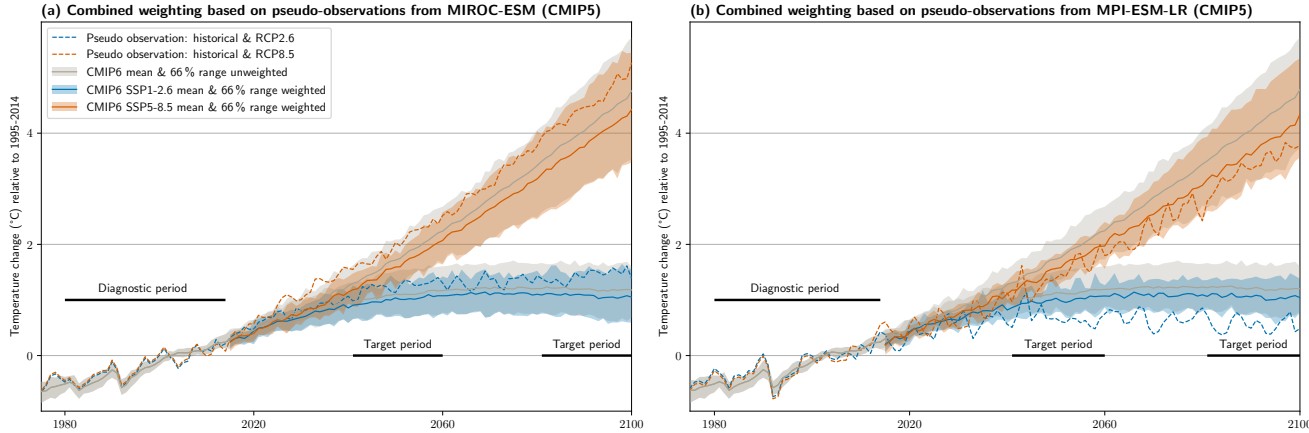

**Figure 2.** Time series of temperature change (relative to 1995-2014) for unweighted (gray) and weighted (colored) CMIP6 mean (lines) and likely (66 %) range (shading) as well as the CMIP5 models serving as pseudo-observations (dashed lines). Shown are the cases which lead to (a) the largest decrease in skill (CMIP5 pseudo-observation: MIROC-ESM) and (b) to the largest increase (MPI-ESM-LR) for SSP5-8.5 in the end-of-century target period. Note that no inference on the performance of the CMIP5 models can be drawn from this figure. Diagnostic period refers to the 1980-2014 period, which informs the weights; the target periods to 2041-2060 and 2081-2100.

case. Since the goal of the weighting is to improve the projections based on performance and dependence of the models, the risk of negative CRPSSs should be minimised.

We find the $\sigma_D$-values to be correctly calibrated by the method in order to limit the risk for a strong skill decrease (CRPSS is close to zero or positive for the 25th percentile in almost all cases). For the mid-century period, the median skill increases by up to 25 % depending on SSP and combination of diagnostics. The magnitude of potential negative CRPSSs in a "worst-case" scenario (5th percentile), however, is better constrained using a balanced combination of diagnostics (e.g., 50 % tasTREND). In the end-of-century period, the median skill is more variable (mainly due to the selected performance shape parameters $\sigma_D$; see table S1), with combinations that include both trend and not-trend diagnostics again performing best.

Using 50 % tasTREND and 50 % anomaly- and variance-based diagnostics (about 13 % tasANOM, 13 % tasSTD, 13 % pslANOM, and 13 % pslSTD) optimises the combination of median CRPSS increases and avoidance of possible negative CRPSSs; we therefore use this combination to calculate the weights for the rest of the analysis. Note that the two SSPs and time periods have slightly different $\sigma_D$ values (ranging from 0.35 to 0.58; table S1), leading to slightly differing weights even though the historical information is the same. This arises from differences in confidence when applying the method for different targets. However, since the $\sigma_D$ values are found to be so similar we use the mean value from the two SSPs and time periods in the following for simplicity, hence $\sigma_D = 0.43$. This does not have a strong influence on the results but simplifies their presentation and interpretation.

## 3.2 Perfect model test using CMIP5 as pseudo-observations

We now use each of the 27 CMIP5 models in turn as pseudo-observation and include both the performance and independence parts of the method. For all considerations in this section, we use the CMIP5 merged historical and RCP runs corresponding to the CMIP6 historical and SSP runs, i.e., RCP2.6 to SSP1-2.6 and RCP8.5 to SSP5-8.5. This allows an evaluation of the skill of the full weighting method applied to the CMIP6 MME in the future. Figure 2 shows two cases selected to lead to the largest decrease (figure 2a) and increase (figure 2b) in the CRPSS for SSP5-8.5 in the end-of-century period when applying the weights. This reveals an important feature of constraining methods in general: there is a risk that the information from the historical period might not lead to a skill increase in the future. In the case shown in figure 2a weighting based on pseudo-observations from MIROC-ESM shifts the distribution downwards, while projections from MIROC-ESM end up warming more than the unweighted mean in the future. This reflects the possibility that information drawn from real historical observations might not lead to an increase in projection skill in some cases. Here cases of decreasing skill appear for about $15\%$ of pseudo-observations.

The largest skill increases, in turn, often comes from pseudo-observations rather far away from the unweighted mean. It seems that, if the pseudo-observations behave very different from the model ensemble in the historical period, there is a good chance that they will continue to do so in the future. One explanation for this could be a systematic difference between the models in the ensemble and the pseudo observation due to, e.g., a missing feedback or component. An important cautionary takeaway is thus to not only maximise mean skill increase when setting up the method, as the cases with highest skill might come from rather "unrealistic" pseudo-observations (i.e., the ones on the tails of the model distribution). This is illustrated in figure S5 in the supplement (e.g., using the CMIP5 GFDL or GISS models as pseudo observations). However, in many cases we do not necessarily expect the real climate to follow such an extreme trajectory but rather be closer to the unweighted MME mean (in part because real observations tend to be used in model development and tuning). It is thus important to use a balanced set of multiple diagnostics and not only optimise for maximal correlation in choosing $\sigma_D$, which might make the highest possible skill increases unattainable, but – maybe more importantly – guard against even more substantial skill decreases.

Finally, it is important to note that the skill of the weighting for a given pseudo-observation also depends on the target. In isolated cases that can mean that the weighting leads to an increase in skill for one SSP while it leads to a decrease in the other (e.g., IPSL-CM5A-LR as pseudo-observation) or to an increase in one time period and to a decrease in the other (e.g., CSIRO-Mk3-6-0). An overview of the weighting based on each of the 27 CMIP5 models can be found in figure S5 in the supplement.

To look into the skill change more quantitatively, figure 3a shows the skill distribution of weighting CMIP6 to predict each of the pseudo-observations drawn from CMIP5 for both target time periods and scenarios. We note again that for each CMIP5 pseudo-observation directly related CMIP6 models are excluded (see table S5 for a list). Compared to the leave-one-out perfect model test with CMIP6 shown in figure 1 the increase in median CRPSS is lower and the risk for negative CRPSSs is slightly higher. This is not unexpected for a test sample which is structurally different from CMIP6 in several aspects (such as forcing

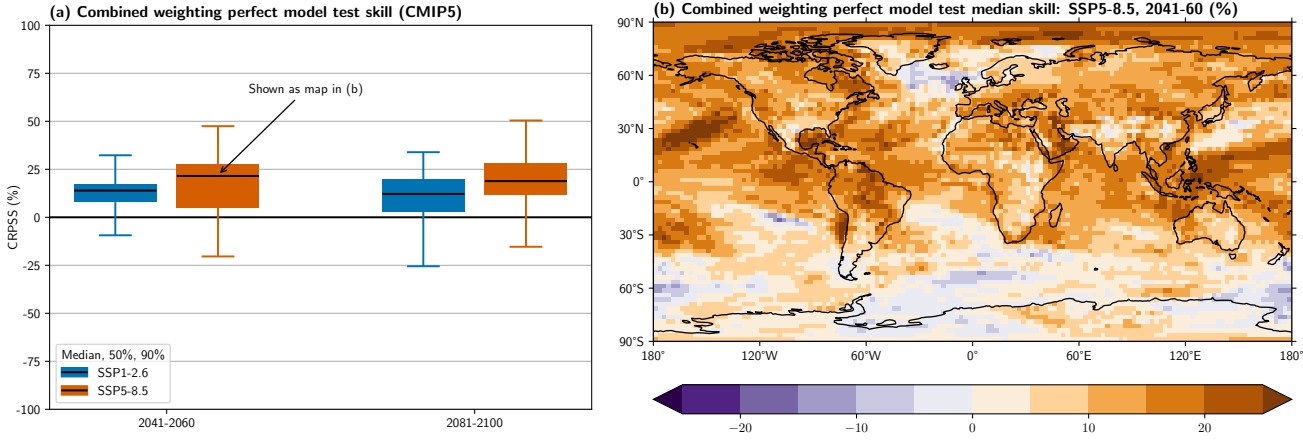

**Figure 3.** (a) Similar to figure 1 but using 27 CMIP5 models as pseudo-observations and showing only the $50\,\%$ tasTREND case. (b) Map of median of the CRPSS relative to the unweighted ensemble for 2041-2060 under SSP5-8.5

320 scheme and maximum amount of warming). But the setup still achieves a median CRPSS increase of about $12\,\%$ to $22\,\%$, with the risk for a skill reduction being confined to about $15\,\%$ of cases and to a maximum decrease of about $25\,\%$. This clearly shows that ClimWIP can be used to provide reliable estimates of future global temperature change and related uncertainties from the CMIP6 MME.

 Finally, we consider the question of whether there are regional patterns in the skill change by investigating a map of median
325 CRPSSs for SSP5-8.5 in the mid-century period in figure 3b (see figure S6 in the supplement for the other cases). Note that each CMIP6 model is still assigned only one weight, but the CRPSS is calculated at each grid point separately. The skill increases almost everywhere with the northern hemisphere having a slightly higher amplitude. A notable exception is the North Atlantic, where weighting leads to a slight decrease in median skill. Indeed, this is the only region where the unweighted CMIP6 mean underestimates the warming from CMIP5. Weighting the CMIP6 ensemble leads to a slight strengthening of the
330 underestimation in this region, while it reduces the difference almost everywhere else.

 In summary, weighting CMIP6 in a perfect model test using five different diagnostics to establish model performance and two diagnostics for independence shows a clear increase in median skill compared to the unweighted distribution consistent over both investigated scenarios and time periods. Looking into the geographical distribution reveals an increase in skill almost everywhere, with some decreases found in the Southern Ocean, particularly in SSP1-2.6 (figure S6). Importantly, skill increases
335 almost everywhere over land, thus benefiting assessments of climate impacts and adaptation where people are affected most directly.

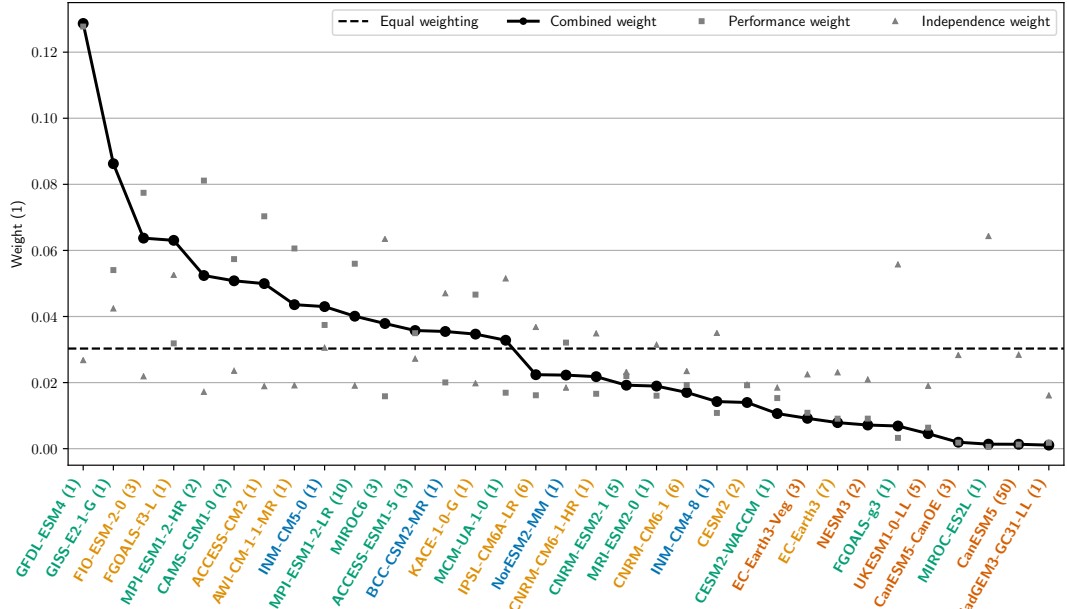

**Figure 4.** Combined independence-performance weights for each CMIP6 model (line with dots) was well as pure performance weights (squares) and pure independence weights (triangles). All three cases are individually normalised and the equal weighting each model would receive in a normal arithmetic mean is shown for reference (dashed line). The labels are coloured by each models TCR value: $> 2.5\,°\mathrm{C}$ - red, $> 2\,°\mathrm{C}$ - yellow, $> 1.5\,°\mathrm{C}$ - green, and $\leq 1.5\,°\mathrm{C}$ - blue. The number of ensemble members per model is shown in brackets after the model name.

## 4   Weighting CMIP6 projections of future warming based on observations

So far we have selected a combination of diagnostics, which leads to the highest increase in median skill while minimising the risk for a skill decrease based on an out-of-sample perfect model test with CMIP6 in section 3.1. We also argued that we use the same shape parameters (which determine the strength of the weighting) for all cases, namely $\sigma_S = 0.54$ for independence and $\sigma_D = 0.43$ for performance. In section 3.2 we then evaluated this setup by using 27 pseudo-observations drawn from the CMIP5 MME. In this section we now calculate weights for CMIP6 based on observed climate and validate the effect of the independence weighting. We use observational surface air temperature and sea level pressure estimates from the ERA5 and MERRA2 reanalyses to calculate the performance diagnostics (tasANOM, tasSTD, tasTREND, pslANOM, pslSTD). As independence diagnostics we continue to use model-model distances in tasCLIM and pslCLIM.

### 4.1   Calculation of weights for CMIP6

Figure 4 shows the combined performance and independence weights assigned to each CMIP6 model by ClimWIP when applied to the target of global temperature change. In addition also the individual performance and independence weights are

shown. All three cases are individually normalised. Applying the combined weight, about half of the models receive more weight than in a simple arithmetic mean and about half receive less. The best performing model, GFDL-ESM4, has about four times more influence than it would have without weighting (about 0.13 compared to 0.03 in the case with equal weighting). The three lowest performing models, MIROC-ES2L, CanESM5, and HadGEM3-GC31-LL, in turn receive less than 1/20 of the equal weighting (about 0.001).

Indeed, several recent studies have found that models which show more future warming per unit of greenhouse gas are less likely based on comparison with past observations (e.g., Jiménez-de-la Cuesta and Mauritsen, 2019; Nijsse et al., 2020; Tokarska et al., 2020). Consistent with their findings models with high TCR receive very low performance (and combined) weights (label colours in figure 4). Among the five lowest ranking models four have a TCR above $2.5\,°C$ and all models with TCR above $2.5\,°C$ receive less then equal weight. The eight highest ranking models, in turn, have TCR values ranging from $1.5\,°C$ to $2.5\,°C$ and lie, therefore, rather in the middle of the CMIP6 TCR range. See table S2 in the supplement for a summary of all model weights and TCR values.

In addition to the combined weighting, figure 4 also shows the independence and performance weights separately. We discuss model independence in more detail in the next section. For the model performance weighting, the relative difference to the combined weighting (i.e., the influence of the independence weighting) is mostly below $50\,\%$, with the MIROC model family being one notable exception. Both MIROC models are very independent, which shifts MIROC6 from a below-average model (based on the pure performance weight; square in figure 4) to an above-average model in the combined weight (dot) effectively more than doubling its performance weight. For MIROC-ES2L the scaling due to independence is similarly high, but its total weight is still dominated by the very low performance weight. In the next section we investigate if these independence weights indeed correctly represent the complex model inter-dependencies in the CMIP6 MME and down-weight models which are highly dependent on other models appropriately.

## 4.2 Validation of the independence weighting

Focusing on the independence weights in figure 4 one can broadly distinguish three cases: (i) relatively independent models, (ii) clusters of models which are quite dependent, and (iii) models for which the independence weighting does not really influence the weighting. To visualise and discuss these cases somewhat quantitatively, we show a CMIP6 model family tree similar to the work by Masson and Knutti (2011) and Knutti et al. (2013).

Using the same two diagnostics, namely horizontally resolved global temperature and sea level pressure climatologies (from 1980-2014) we apply a hierarchical clustering approach (section 2.7). Figure 5 shows the resulting family tree of CMIP6 models similar to the work by Masson and Knutti (2011) and Knutti et al. (2013). In this tree models which are closely related branch further to the left, while very independent model clusters branch further to the right. The mean distance between two initial-condition members of the same as an estimation for the internal variability in the generalised distance is indicated as grey shading. Model which have a distance similar to this value (e.g., the two CanESM5 model versions) are basically indistinguishable. The independence shape parameter used through the manuscript ($\sigma_S = 0.54$) is shown as dashed vertical line.

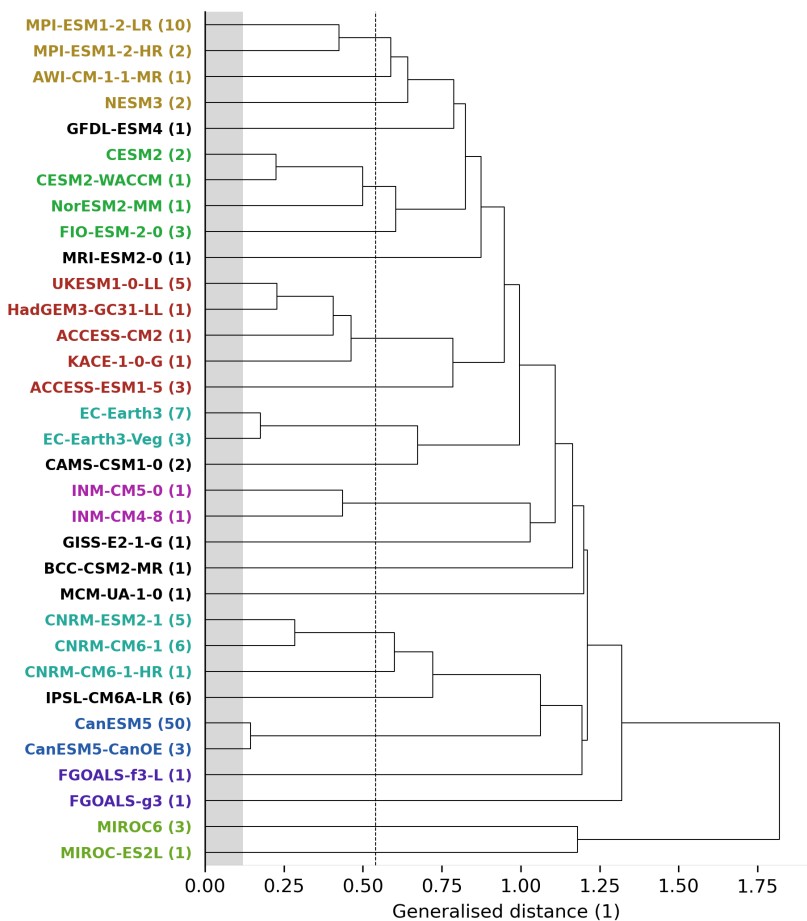

**Figure 5.** Model family tree for all 33 CMIP6 models used in this study similar to Knutti et al. (2013). Models branching further to the left are more dependent, models branching further to the right are more independent. The analysis is based on global, horizontally resolved tasCLIM and pslCLIM in the period 1980-2014. The independence shape parameter $\sigma_S$ is indicated as dashed vertical line, an estimation of internal variability as grey shading. Labels with the same colour indicate models with obvious dependencies such as shared components or same origin (models with no clear dependencies are labelled in black).

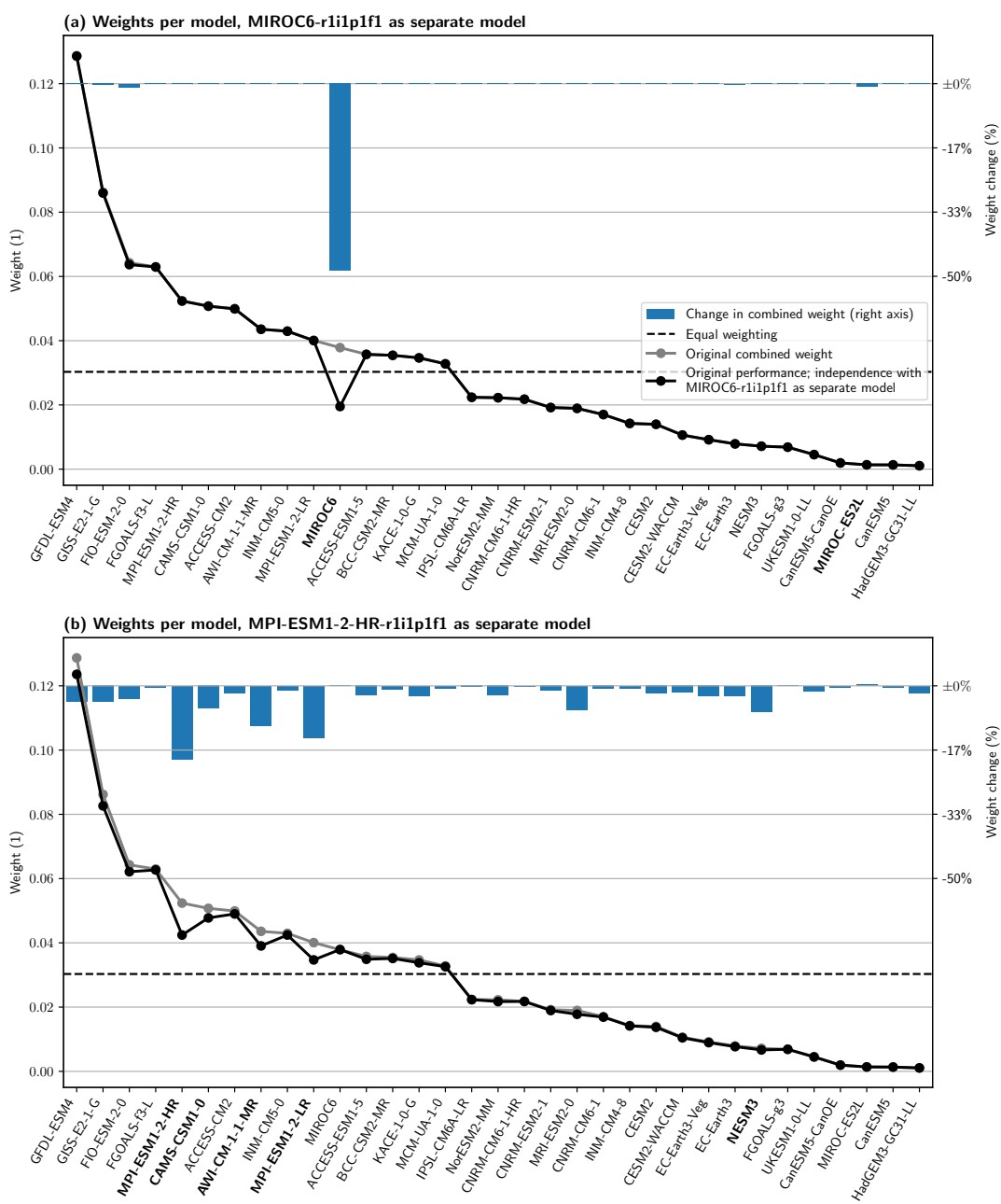

**Figure 6.** Similar to figure 4 but removing one initial-condition ensemble member from (a) MIROC6 and (b) MPI-ESM1-2-HR and adding it as separate model when calculating the independence weights (the "new" model is not shown in the plot). Models with obvious dependencies to the "new" model have bold labels (equivalent to figure 5). The change in the combined weight relative to the original weight is shown as blue bars using the right axis.

A comprehensive investigation of the complex inter-dependencies within the multi-model ensemble in use and further between models from the same institution or of similar origin is beyond the scope of this study and will be subject of future work.

Here we limit ourselves to pointing out several base features of the output-based clustering, which serve as indications that it is skilful in identifying inter-dependent models. The labels of models with the same origin or with known shared components are marked in the same colour in figure 5. These two factors are the most objective measure for a priori model dependence we have. The information about the model components is taken from each model's description page on the ES-DOC explorer (https://es-doc.org/cmip6/) as listed in table S4 in the supplement.

Figure 5 clearly shows that clustering models based on the selected diagnostics performs well: models with shared components or with the same origin (indicated by the same colour) are always grouped together. Looking into a bit more detail we find, for example, that closely related models such as low and high resolution versions (MPI-ESM-2-LR and MPI-ESM-2-HR; CNRM-CM6-1 and CNRM-CM6-1-HR) or versions with only one differing component (CESM2 and CESM2-WACCM; INM-CM5-0 and INM-CM4-8; both differing only in the atmosphere) are detected as being very similar. Both MIROC models,

which have been identified as very independent based on figure 4, in turn, are found to be very far away from each other and even further away from all other models in the CMIP6 MME.

To investigate if the independence weighting correctly translates model distance into weights we now look at two models as examples: one that performs well and is relatively independent (MIROC6) and another that also performs well but is more dependent (MPI-ESM1-2-HR). Each has multiple ensemble members; we remove one member from each and add it to the

MME as an additional model as detailed in section 2.7.

In the first case (figure 6a; MIROC6 which is among the least dependent models), the original weight is reduced by almost 1/2, which is close to what we would expect in the idealised case. All other models are unaffected by adding a duplicate of MIROC6, even the other model from the same center, MIROC-ES2L which differs in atmospheric resolution and cumulus treatment (Tatebe et al., 2019; Hajima et al., 2019). Based on the "family tree" shown in figure 5 this behaviour is not surprising:

the two MIROC models are not only identified as the most independent models in the CMIP6 MME but also as very independent from each other. While some of the components and parameterizations are similar, updates in parameterizations and in the tuning of the parameters appear to be sufficient here to create a model that behaves quite differently.

The second case (figure 6b; MPI-ESM1-2-HR which is among the most dependent models) shows a very different picture. The strongest effect on the original weight is found for the copied model itself, which is reduced by about $20\,\%$, but also several

other models are affected. Looking into these models in more detail, we conclude that the inter-dependencies detected by our method can be traced to shared components in most cases: MPI-ESM1-2-LR is just the low resolution version of MPI-ESM1-2-HR (run with a T63 atmosphere instead of T127 and a $1.5°$ ocean instead of $0.4°$), AWI-CM-1-1-MR and NESM3 share the atmospheric (ECHAM6.3) and similar land (JSBACH3.x) components, and CAMS-CSM1-0 shares a similar atmospheric (ECHAM5) component, while MRI-ESM2-0 does not have any obvious dependencies. Information about the models can be

found in their reference publications (Mauritsen et al., 2019; Gutjahr et al., 2019; Semmler et al., 2019; Yang et al., 2020; Chen et al., 2019; Yukimoto et al., 2019) and on the ES-DOC explorer, which provides detailed information about all model used in this study. The links to each models information page can be found in table S4 in the supplementary material.

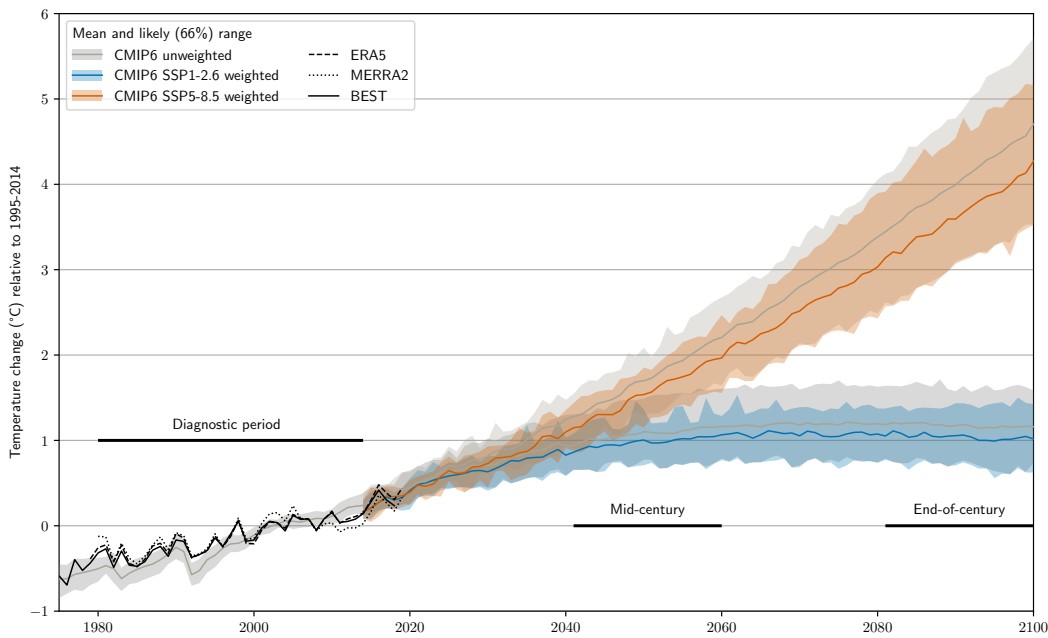

**Figure 7.** Timeseries of temperature change (relative to 1995-2014) for unweighted (gray) and weighted (colored) CMIP6 mean (lines) and likely (66 %) range (shading). Three observational datasets are also shown in black; note that BEST is not used to inform the weighting and is only shown for comparison here.

## 4.3    Applying weights to CMIP6 temperature projections and TCR

Figure 7 shows a timeseries of unweighted and weighted projections based on a weak (SSP1-2.6) and strong (SSP5-8.5) climate
change scenario. For both scenarios a clear shift in the mean towards less warming is visible, which is also reflected in the upper uncertainty bound. Notably, however, the lower bound hardly changes, leading to a reduction in projection uncertainty in total. This becomes even clearer when investigating the two 20-year periods, reflecting mid- and end-of-century conditions (figure 8a and table S3).

Based on these results, warming exceeding $5\,°C$ by the end of the century is very unlikely even under the strongest climate
change scenario SSP5-8.5. The mean warming for this case is shifted downward to about $3.7\,°C$ and the 66 % (likely) and 90 % ranges are reduced by 12 % and 30 %, respectively. For SSP1-2.6 in the end-of-century period as well as both SSPs in the mid-century period, reductions in the mean warming of $0.1\,°C$ to $0.2\,°C$ are found. The likely range is reduced by about 20 % to 30 % in these three cases. A summary of weights and warming values for all models as well as all statistics can be found in tables S2 and S3 in the supplement. Recent studies that use historical temperature trend as an observational constraint for
future warming lead to similar conclusions, with lower constrained warming compared to unconstrained (both in the mean and upper percentiles of the distributions) (e.g., Nijsse et al., 2020; Tokarska et al., 2020).

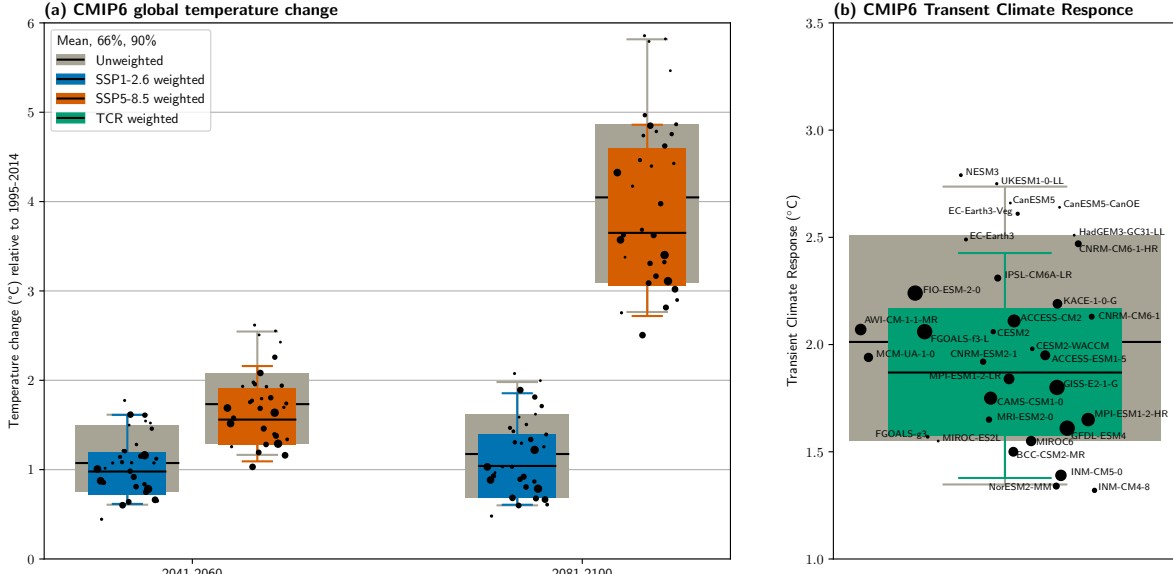

**Figure 8.** (a) Unweighted (gray) and weighted (colors) temperature change (relative to 1995-2014) for both periods and scenarios. (b) Unweighted (gray) and weighted (green) Transient Climate Response (TCR). The dots show individual models as labelled, with the dot size indicating the weight. The horizontal dot position is arbitrary.

To investigate the influence of remaining internal variability in our combination of diagnostics on the weighting, we also perform a bootstrap test. Selecting only one random member per model (for models with more than one ensemble member) we calculate weights and the corresponding unweighted and weighted temperature change distributions. This is repeated 100

times, providing uncertainty estimates for both the unweighted and weighted percentiles. The mean values of the weighted percentiles taken over all 100 bootstrap samples are very similar to the values from the weighting based on the full MME (including all ensemble members; see figure S7) confirming the robustness of our approach.

We also apply weights to TCR estimates in figure 8b, finding an unweighted mean TCR value of about 2 °C with a likely range of 1.6 °C to 2.5 °C. Weighting by historical model performance and independence constrains this to 1.9 °C (1.6 °C to

2.2 °C), a reduction of 36 % in the likely range. These values are consistent with recent studies based on emergent constraints which estimate the likely range of TCR to be 1.3 °C to 2.1 °C (Nijsse et al., 2020) and 1.2 °C to 2.0 °C (Tokarska et al., 2020) and they are also very similar to the range of 1.5° to 2.2° from Sherwood et al. (2020) who combine multiple lines of evidence. They are also consistent but substantially more narrow than the likely range from the fifth assessment report of the IPCC (IPCC, 2013) based on CMIP5: 1 °C to 2.5 °C. Figure 8b clearly shows that almost all models with higher than equal weights lie within

the likely range, and only one model lies above it (FIO-ESM-2-0). This is a strong indication that TCR values beyond about 2.5 °C are unlikely when weighting based on several diagnostics and when accounting for model independence.

## 5 Discussion and Conclusions

We have used the Climate model Weighting by Independence and Performance (ClimWIP) method to constrain projections of future global temperature change from the CMIP6 multi-model ensemble. Based on a leave-one-out perfect model test, a combination of five global, horizontally-resolved diagnostic fields (anomaly, variance, and trend of surface air temperature and anomaly and variance of sea level pressure) was selected to inform the performance weighting. The skill of weighting based on this selection was tested and confirmed in a second perfect model test using CMIP5 models as pseudo-observations. Our results clearly show the usefulness of this weighting approach in translating model spread into reliable estimates of future changes and in particular into uncertainties that are consistent with observations of present day climate and observed trends.

We also discussed the remaining risk for decreasing skill compared to the raw distribution which is a crucial question in all weighting or constraining methods. We show the importance of using a balanced combination of climate system features (i.e., diagnostics) relevant for the target to inform the weighting to minimise the risk for skill decreases. This guards against the possibility of a model "accidentally" fitting observations for a single diagnostic while being far away from them in several others (and hence possibly not providing a skilful projection of the target variable).

By adding copies of existing models into the CMIP6 multi-model ensemble we verified the effect of the independence weighting, showing that models get correctly down-weighted based on an estimate of dependence derived from their output. To inform the independence weighting we used two global, horizontally resolved fields (climatology of surface air temperature and sea level pressure) which we showed to allow a clear clustering of models with obvious inter-dependencies using a CMIP6 "family tree".

From these tests we conclude that ClimWIP is skilful in weighting global mean temperature change from CMIP6 using the selected setup. We hence use it to calculate weights for each CMIP6 model and apply them in order to obtain probabilistic estimates of future changes. Compared to the unweighted case these results clearly show that the CMIP6 models which lead to the highest warming are less probable, confirming earlier studies (e.g., Nijsse et al., 2020; Sherwood et al., 2020; Tokarska et al., 2020). We find a weighted mean global temperature change (relative to 1995-2014) of $3.7\,°C$ with a likely (66 %) range of $3.1\,°C$ to $4.6\,°C$ by the end of the century when following SSP5-8.5. With ambitious climate mitigation (SSP1-2.6) a weighted mean change of $1\,°C$ (likely range: $0.7\,°C$ to $1.4\,°C$) is projected for the same period.

On the policy level, this highlights the need for quick and decisive climate action to achieve the Paris climate targets. For climate modeling on the other hand, this approach demonstrates the potential to narrow the uncertainties in CMIP6 projections, particular on the upper bound. The large investments in climate model development have so far not led to reduced model spread in the raw ensemble, but the use of climatological information and emergent transient constraints has the potential to provide more robust projections with reduced uncertainties, that at the same time are more consistent with observed trends, thus maximizing the value of climate model information for impacts and adaptation.

*Code availability.* The ClimWIP model weighting package is available under a GPLv3 at https://github.com/lukasbrunner/ClimWIP.git

*Author contributions.* LB, ALM, and RK were involved in conceiving the study. LB did the analysis and created the plots substantially supported by AGP. LB wrote the manuscript with contributions from all authors. The ClimWIP package was implemented by LB and RL; AGP wrote the script used to create tables S4 and S6.

*Competing interests.* The authors declare that they have no conflict of interest.

*Acknowledgements.* The authors thank Martin B. Stolpe for providing the TCR values as well as Martin B. Stolpe and Katarzyna B. Tokarska for helpful discussions and comments on the manuscript. This work was carried out in the frame of the EUCP project which is funded by the European Commission through the Horizon 2020 Programme for Research and Innovation: Grant Agreement 776613. Ruth Lorenz was funded and Anna L. Merrifield was co-funded by the European Union's Horizon 2020 Research and Innovation program: Grant Agreement 641816 (CRESCENDO). Flavio Lehner was supported by a Swiss NSF Ambizione Fellowship (Project PZ00P2_174128). This material is partly based upon work supported by the National Center for Atmospheric Research, which is a major facility sponsored by the National Science Foundation (NSF) under Cooperative Agreement No. 1947282, and by the Regional and Global Model Analysis (RGMA) component of the Earth and Environmental System Modeling Program of the U.S. Department of Energy's Office of Biological & Environmental Research (BER) via NSF IA 1844590. This study was generated using Copernicus Climate Change Service Information 2020 from ERA5. The authors thank NASA for providing MERRA2 and Berkeley Earth for providing BEST. We acknowledge the World Climate Research Programme, which, through its Working Group on Coupled Modelling, coordinated and promoted CMIP5 and 6. We thank the climate modeling groups for producing and making available their model output, the Earth System Grid Federation (ESGF) for archiving the data and providing access, and the multiple funding agencies who support CMIP5 and 6 and ESGF. A list of all CMIP6 runs and their references can be found in table S6 in the supplement. We thank all contributors to the numerous open source packages which were crucial for this work, in particular the Python project xarray (http://xarray.pydata.org). The authors thank two anonymous reviewers for their helpful comments on our work.

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
