# Peer review of "Reduced global warming from CMIP6 projections when weighting models by performance and independence"

_Earth System Dynamics, 2020_

## Referee Comment (RC1) · Anonymous Referee #1 · 3 Jun 2020

Summary The authors present a methodology for weighting CMIP6 models based on several performance metrics as well as on their independence from each other. This provides narrower bounds on future global mean temperature changes than in the unweighted ensemble, primarily by down-weighting the highly sensitive models that happen to have poor performance with respect to two reanalysis products and/or are closely related to other models. I found the paper to be nicely motivated, well organized and supported, and a useful contribution to the literature. There are a few areas that I think need to be clarified, and so I recommend minor revisions.

Major Comments

\* Figure 1 and the discussion around lines 241-242: the terminology of 0% to 100% trend-based seems too ambiguous to me and should just be written out explicitly. Couldn't the terms that are included just be stated explicitly in the figure? The figure doesn't really stand on its own, since one has to refer to these lines to know what exactly is meant by these. Additionally, it is not clear what the intermediate values (33%, 50%, 66%) correspond to exactly. Upon multiple readings, I still cannot understand what is meant by these percentages at all, and I'm not completely sure what is actually meant by "50% tasTREND and 50% anomaly- and variance-based diagnostics" that forms the basis of the remaining analysis. Please clarify.

\* Discussion of Figure 2 around line 270: Should one have intuitively expected this from the math? I cannot seem to rationalize why using a model that is close to the CMIP6 MME to weigh CMIP6 would pull the CMIP6 MME mean away from the pseudo-observational "truth". This seems like a deficiency in the weighting. Shouldn't the weighting be resilient to this and do very little "harm" in this case?

\* Figure 4: The combined and performance-only weights are shown, but not the independence weights. Is there a reason for this? Is it worth also showing the ECS or TCR from these models on this plot, so that one could see that higher ECS/TCR models tend to be down-weighted? I assume this is correct, to the extent that models that warm the most over the 21st Century have high ECS/TCR, but I don't recall the authors coming out and saying it. Modifying this figure in this way could be a compact way of making that point.

\* Figure 4: I'm surprised to see several well-regarded models having relatively low performance weights (UKESM, HadGEM, CanESM, CESM), whereas some models that are typically poor performers seem to do well here (GISS, FGOALS, INM-CM). Any comment? Is it possible that your performance metrics are too restrictive (just involving tas and psl, two fields that may not adequately discriminate models with good vs bad moist physics that governs feedback and ECS), allowing poor performing models to get high weights?

[Figure]

Minor Comments

\*line 61: should be "model's"

\*line 78: should be "method's"

\*Line 250: I don't see where the 10-20% statement comes from. By my eye, the medians range from near 0% to slightly larger than 25%.

\*Figure 1: titles should be "leave-one-out"

\*Figure 2 caption: should be "which"

\*Figure 2: To clarify, the similarity between pseudo-obs and MME is only assessed over the "Diagnostic period" right? (Side-note: "diagnostic period" only appears in the figure and is not discussed in the text.) By my eye, MPI looks closer to the MME than does CanESM, so I'm a bit confused here. Is the reason because similarity in the evolution of GMST only one of the several metrics employed, and MPI does worse in the ones that cannot be gleaned from this figure?

\*Line 309: "allows us" or "allows one"; also, it seems like some reference to all the performance metrics work done by Gleckler et al seems appropriate here. I believe they also advocate for comparing against multiple observational datasets.

\*Line 314: I don't see the motivation for these 3 groupings. Is it in any way objective?

\*Figure 6: too small to read, suggest stacking the two panels vertically rather than placing them next to each other horizontally

\*Line 334: should be "model's"

\*I don't think the average reader should be expected to know how to interpret a figure like Figure 5. Only the meaning of the colors are explained in the caption. What does the rest signify?

\*Line 391 "The weighting also largely reconciles CMIP6 with 5": what is this referring

to specifically, and is there a figure in particular being referenced?

*Figure 4: Are all weights less than or equal to 1 in absolute units, and only exceed 1 when expressed relative to equal weighting as is done in the figure? Otherwise I'm a little confused about why a model would have a weight in excess of 1. How exactly is wi used? weighted avg of X = sum(wi*Xi)/sum(wi)?

---

## Referee Comment (RC2) · Anonymous Referee #2 · 23 Jul 2020

General Comments

Some models are more consistent with historical observations than others. In climate projection, it makes intuitive sense to give more weight to the models that are more consistent with observed climate shifts and less weight to models that are less consistent with observed climate shifts.

But how?

This paper reports on a method of assigning model weights that relies on two distinct distance measures: the distance of models from observations and the distance of models from other models. The method requires the specification of two parameters

that determine how each of these distances are turned into model weights. The method for determining the parameter associated with inter-model distance is poorly explained (see specific comment 8 below). The method for determining the parameter associated with distance from observations is also poorly explained, but for many experiments, involves future-time-pseudo-observations from the future states that are the objective of the prediction (see comment 10 below). In other words, the tuning method appears to render the tests of the method to be of the "in-sample" variety. To weaken the degree of "in-sampleness" an additional test is performed using CMIP5 runs. However, since one expects many of the CMIP6 models to be closely related to the CMIP5 models, there are strong reasons to believe that this test is not truly "out-of-sample" either.

Even with the use of "future-time-pseudo-observations" in the tuning procedure, the improvements from this weighting scheme seem very modest in comparison with, for example, those obtained in Abramowitz and Bishop (2015, J. Climate) – (using a a method that solely required historical observations for the weights). The revised paper should include some attempt to compare/contrast/explain the Abramowitz and Bishop results.

A superficially appealing feature of the method is that it gives more weight to models that are both skillful and statistically independent of other models. However, this independence is just described in terms of inter-model distance and not in terms of the independence of the model error. Is there some unstated proof that increased inter-model distance equates to increased model error independence? (It seems easy to think of counter examples). As demonstrated in Bishop and Abramowitz (2013), it is the independence of the error of the individual models comprising an ensemble forecast (as measured by inter-model forecast error correlation) that increases the predictive power of the ensemble. The revised paper needs to address the issue of the relationship or lack of relationship between inter-model distance and model error independence.

After applying the method to the CMIP6 ensemble members, the authors find reduced

warming relative to the simple sample means of CMIP6 ensembles for the high and low $CO_2$ concentration scenarios considered. However, any confidence in this prediction must be strongly tempered by the "in sample" circular- nature of the testing and tuning procedures used by this method.

My overall recommendation would be that the paper be returned to the authors to address the specific comments below and to include results from experiments in which only historical observations (or model-based-historical-pseudo-observations) were used to determine the weights. This constitutes major revision.

Specific Comments

1. Line 16. Consider explaining what TCR is in the abstract to appeal to a broader audience.

2. Line 31. Do you mean model uncertainty, unknown model climate error, unknown model-climate-sensitivity-to-CO2 error or model climate differences? We know what the model is, and we can determine its climate past, present and future by running it. We can also determine the differences between the climates of different models. Given the limitations of the spatio-temporal distribution of observations, the uncertain thing is the actual climate both past, present, and future, is it not?

3. Line 35. Lorenz, the father of chaos theory, argued that while the accuracy of weather forecasts was limited to a few weeks the climate of a system was not sensitive to specified initial conditions and could be known provided the forcing on the system was known. I guess "climate" in the sense of Lorenz refers to the statistical description of the attractor of the chaotic system. When you refer to "internal variability" do you just mean slow modes of the model's chaotic attractor that might possibly be confused with a change in the mean of the model's attractor if the ensemble size was too small?

4. Line 102: I'm guessing you are referring to Section 3.2 of Brunner et al., 2019. Is that correct? If so, please state this in the text. Your wording suggested that you had

estimated an observation error variance. However, on reading Section 3.2 of Brunner et al., 2019, I'm now guessing that you are referring to how your derived weights change depending on which subset of all observations you use. Are you suggesting that the reason for your weights changing is because the observations have different errors? Can you rule out the possibility that your weighting scheme isn't just over-fitting each individual observational data set? In any case, the revised paper needs to clarify whether in fact you are referring to the size of the change in weights associated with using differing observational data sets. Also, the observed values are known. They are not uncertain. The errors of the observed values are unknown. It is the observational error that is uncertain.

5. Line 145. "We want to . . ." If there was a hypothetical user of the climate projection that only cared about temperature trend and not about year-to-year variability, might you not be doing them a disservice by down-weighting members that have an excellent temperature trend but poor inter-annual variability? Consider changing to "We choose to . . . "

6. Line 147-149. Equations should be added to precisely describe these observation derived quantities – perhaps in an appendix or supplementary material.

7. Line 170. You must state what was used as a proxy for a perfect model. I would think that the derived sigma_D must be related to the ensemble variance of the model states around the time averaged state. That quantity will depend on the model will it not? Please clarify.

8. Line 183. I looked at Section 2.3 of Brunner et al., 2019 for an explanation but Brunner et al. (2019) just directs the reader to Lorenz et al., 2018. Your work needs to be reproducible. When referring to another paper for a key explanation, you must give very specific information about where in the paper the explanation resides (e.g. a section number) to ensure reproducibility. You have not done this.

9. Line 191. The method used to evaluate performance given here seems almost

identical to that given in Abramowitz and Bishop (2015) but no reference is given to this paper or others that may have used this approach before. Such literature is relevant and should be cited.

10. Line 200-205. Here, we learn that sigma_D weights are determined in part from information from a place that is inaccessible in reality: the future. Only model futures are accessible. By line 205 we learn that the model future states (rather than observations) are, in fact, an integral part of choosing the weights. This is a significant departure from many other observation-based methods for improving ensemble forecasts and projections. The use of future time observations in the training causes all of the associated tests to be "in-sample" tests – dramatically reducing their trustworthiness. Since the CMIP5 models belong to the same general class of human produced climate simulators they can barely be considered "out-of-sample". Please comment on the limitations of this approach. In addition, you have not clarified how the method of tuning for future states interacts with the method to determine sigma_D referred to on line 170 (see previous comment).

11. Line 266-280. Here we learn that the method is very prone to creating decreased skill relative to the multi-model unweighted mean. This negative result is in contrast to the positive results found in Abramowitz and Bishop (2015) using the method of Bishop and Abramowitz (2013).

---

## Author Comment (AC1) · 20 Sep 2020

Please see the attached 'Brunner_etal_answers_reviewer1.pdf' for our answers.

Please also note the supplement to this comment:
https://esd.copernicus.org/preprints/esd-2020-23/esd-2020-23-AC1-supplement.zip

---

## Author Comment (AC2) · 20 Sep 2020

Please see the attached 'Brunner_etal_answers_reviewer2.pdf' for our answers.

Please also note the supplement to this comment:
https://esd.copernicus.org/preprints/esd-2020-23/esd-2020-23-AC2-supplement.zip

---

## Author Response (AR1)

**Editor**

The authors have done an excellent job of responding to the reviewer comments. I am looking forward to seeing the final version of the revised manuscript.

Reviewer #2 raised some valuable points about the chosen metrics in this study and how those metrics compare with others previously used. The authors have done a great job of responding to the reviewer. I would encourage the authors to include some of this information (perhaps focusing on the general response and the response to comment #10) in the supplemental material of their revised manuscript. These sorts of discussions are useful for future studies, and I would hate to see that information get lost.

**We thank the editor for the positive assessment of our manuscript and our responses to the reviewers comments. We agree that it would be potentially helpful to have some of our answers on model independence and the skill tests documented in the supplement. We have, therefore, extended section S3 in the supplement of the revised paper to also include a discussion about the potential circularity between calibration of the performance shape parameter and the subsequent skill tests, which both draw on future model information. For a summary on the different approaches used to establish model independence we have added a new section to the supplement (section S4 in the revised manuscript), where we also show an alternative "family tree" clustering based on the model error correlation distance.**

**Reviewer 1**

Summary

The authors present a methodology for weighting CMIP6 models based on several performance metrics as well as on their independence from each other. This provides narrower bounds on future global mean temperature changes than in the unweighted ensemble, primarily by down-weighting the highly sensitive models that happen to have poor performance with respect to two reanalysis products and/or are closely related to other models. I found the paper to be nicely motivated, well organized and supported, and a useful contribution to the literature. There are a few areas that I think need to be clarified, and so I recommend minor revisions.

**We thank the reviewer for the positive assessment and for the comments on our paper. Please find our answers to the comments highlighted in bold below.**

Major Comments

* Figure 1 and the discussion around lines 241-242: the terminology of 0% to 100%trend-based seems too ambiguous to me and should just be written out explicitly. Couldn't the terms that are included just be stated explicitly in the figure? The figure doesn't really stand on its own, since one has to refer to these lines to know what exactly is meant by these. Additionally, it is not clear what the intermediate values (33%,50%, 66%) correspond to exactly. Upon multiple readings, I still cannot understand what is meant by these percentages at all, and I'm not completely sure what is actually meant by "50% tasTREND and 50% anomaly- and variance-based diagnostics" that forms the basis of the remaining analysis. Please clarify.

**Thank you for pointing this out. The reviewer is correct, our notation in the original manuscript was ambiguous. What we are doing in our analysis is splitting 5 diagnostics into two parts: 1) tasTREND, 2) tasANOM, tasSTD, pslANOM, pslSTD. Each of the categories in figure 1 relates to the relative importance of tasTREND compared to the other diagnostics, i.e.:**

- **0% tasTREND + (25% tasANOM + 25% tasSTD + 25% pslANOM + 25% pslSTD) [termed 'not-trend based' in the manuscript]**

- **33% tasTREND + (17% tasANOM + 17% tasSTD + 17% pslANOM + 17% pslSTD)**

- **50% tasTREND + (13% tasANOM + 13% tasSTD + 13% pslANOM + 13% pslSTD)**

- **66% tasTREND + (8% tasANOM + 8% tasSTD + 8% pslANOM + 8% pslSTD)**

- **100% tasTREND + (0% tasANOM + 0% tasSTD + 0% pslANOM + 0% pslSTD) [termed 'only tasTREND based' in the manuscript]**

**(values not summing up to 100% is due to rounding)**

**We have adjusted the paragraph in question as well as figure 1 in order to make this clearer (see figure 1 and line 259f in the revised manuscript).**

* Discussion of Figure 2 around line 270: Should one have intuitively expected this from the math? I cannot seem to rationalize why using a model that is close to the CMIP6 MME to weigh CMIP6 would pull the CMIP6 MME mean away from the pseudo-observational "truth". This seems like a deficiency in the weighting. Shouldn't the weighting be resilient to this and do very little "harm" in this case?

**Again, thank you for pointing this out. We did not mean to say that cases in which the perfect model is close to the unweighted MME *necessarily* lead to a decrease in skill and there are several examples where this is not the case (e.g., for pseudo observations from CanESM2 or**

IPSL-CM5A-MR; see figure S2 in the revised manuscript). It is crucial, however, to point out that when we write 'close to the truth' we mean close to the truth in the evaluation periods (2041-60 or 2081-00). These periods are not used to inform the weighting and it is possible (in a pure model world as well as in the real world) that the information drawn from the past does not lead to a skill increase in the future if the constraint from the past is unrelated to the future projection. We have adapted our discussion of this topic to be clearer (see lines 291-308 in the revised manuscript).

In addition, skill might be dependent on the emission path. Looking at the time series plots using IPSL-CM5A-LR as pseudo-observations (figure S2 in the revised manuscript), for example, we see a slight downward shift of the distributions for SSP1-2.6 as well as SSP5-8.5. For the former, this leads to an increase in skill while it reduces skill for the latter. We have added a short discussion on this topic to the revised manuscript in lines 309-313 .

We have also added additional information about the skill for each CMIP5 model used as pseudo-observation to figure S2 in the revised manuscript. Finally, we note that figures 2, 4, S2, and S4 have been updated in accordance with a comment from reviewer 2 (see last paragraph of our answer to their comment 10). For each CMIP5 pseudo-observation we now exclude the direct CMIP6 predecessors (if existing) from the calculation (see line 236-237 and table S5 in the revised manuscript).

* Figure 4: The combined and performance-only weights are shown, but not the in-dependence weights. Is there a reason for this? Is it worth also showing the ECS or TCR from these models on this plot, so that one could see that higher ECS/TCR models tend to be down-weighted? I assume this is correct, to the extent that models that warm the most over the 21st Century have high ECS/TCR, but I don't recall the authors coming out and saying it. Modifying this figure in this way could be a compact way of making that point.

We had originally decided against showing independence weights to avoid the readers being overwhelmed by the figure (and because they could be inferred from the difference between combined weights and performance weights). Also, in the original figure we had shown the weights relative to the median weight, so that the distance of a model with, e.g., twice the equal weight would show at the same distance from '1' (equal weighting) as a model with ½ of the weight (see also your last minor comment). However, we realise that this might be slightly harder to interpret so we have changed it in the revised manuscript.

We now show normalised weights for all three cases: independence, performance, and combined. In addition we now indicate TCR by coloring the labels accordingly (Figure 4 in the revised manuscript) and we have added a table containing all values to the supplement (Table S2).

* Figure 4: I'm surprised to see several well-regarded models having relatively low performance weights (UKESM, HadGEM, CanESM, CESM), whereas some models that are typically poor performers seem to do well here (GISS, FGOALS, INM-CM). Any comment? Is it possible that your performance metrics are too restrictive (just involving tas and psl, two fields that may not adequately discriminate models with good vs bad moist physics that governs feedback and ECS), allowing poor performing models to get high weights?

The reviewer is right, several typically well-regarded models receive rather low weights in our scheme. However, we point out that most of the models mentioned as examples have very high TCR. Based on our analysis (and other studies, see, e.g., Tokarska et al., 2020, Nijsse et al., 2020) these very high warming models are less likely and therefore they are down-weighted. In some cases (UKESM, HadGEM, CanESM) the main reason is the obvious mismatch between the observed and simulated warming over the course of the 20th century, which the modeling groups acknowledge in their technical description papers of the models.

It is indeed possible that our particular diagnostics choice leads to typically less well-regarded models receiving relatively high weights. This means that according to our chosen diagnostics

they are performing well compared to other models. It is possible that we would need to include more or other diagnostics to downweight models which have, e.g., bad moist physics, since the weighting method does not include knowledge about specific parameterizations. This point highlights the importance of careful diagnostics choices and the fact that the weighting is always aimed at a particular target and diagnostics choice. The weighting is not supposed to tease out which model is best in every case, and depending on the target and diagnostics choice the models receiving the highest or lowest weights will be different. This does not mean models receiving low weights in this case are bad models in general, as the reviewer realized some low weight models in our case are well regarded models and considered good models in general. But it means that based on their performance in simulating historical warming trends they are considered less likely here.

Minor Comments

*line 61: should be "model's"*line 78: should be "method's"

**Done.**

*Line 250: I don't see where the 10-20% statement comes from. By my eye, the medians range from near 0% to slightly larger than 25%.

**The reviewer is correct, we changed this.**

*Figure 1: titles should be "leave-one-out"

**We changed the caption so this is no longer applicable.**

*Figure 2 caption: should be "which"

**Done.**

*Figure 2: To clarify, the similarity between pseudo-obs and MME is only assessed over the "Diagnostic period" right? (Side-note: "diagnostic period" only appears in the figure and is not discussed in the text.) By my eye, MPI looks closer to the MME than does CanESM, so I'm a bit confused here. Is the reason because similarity in the evolution of GMST only one of the several metrics employed, and MPI does worse in the ones that cannot be gleaned from this figure?

**We now introduce the terms diagnostic period in the main text of the revised manuscript (lines 215). Regarding the second point: the reviewer is correct in assuming that the performance of the models in the diagnostics that inform the weighting can not be inferred from figure 2 in general. We have added a sentence to the caption of figure 2 to make that clear.**

*Line 309: "allows us" or "allows one"; also, it seems like some reference to all the performance metrics work done by Gleckler et al seems appropriate here. I believe they also advocate for comparing against multiple observational datasets.

**This sentence does no longer exist but we have added a reference to Gleckler et al. (2008) in line 108 in the revised manuscript, where we motivate the usage of more than one observational dataset.**

*Line 314: I don't see the motivation for these 3 groupings. Is it in any way objective?

**This paragraph no longer exists in the revised manuscript.**

*Figure 6: too small to read, suggest stacking the two panels vertically rather than placing them next to each other horizontally

**Done.**

*Line 334: should be "model's"

**Done.**

*I don't think the average reader should be expected to know how to interpret a figure like Figure 5. Only the meaning of the colors are explained in the caption. What does the rest signify?

**We have added additional description to figure 5 and now provide a more detailed description of the clustering approach in the supplement (section S6 in the revised manuscript).**

*Line 391 "The weighting also largely reconciles CMIP6 with 5": what is this referring to specifically, and is there a figure in particular being referenced?

**We were referring to the fact that the constrained CMIP6 TCR is closer to the CMIP5 TCR range from, e.g., the IPCC AR5 (1°C-2.5°C). However, this sentence was slightly misplaced here and is no longer included in the revised manuscript.**

*Figure 4: Are all weights less than or equal to 1 in absolute units, and only exceed when expressed relative to equal weighting as is done in the figure? Otherwise I'm a little confused about why a model would have a weight in excess of 1. How exactly is wi used? weighted avg of X = sum(wi*Xi)/sum(wi)?

**We now show normalised weights for all three cases: independence, performance, and combined. See also our answer to your major point regarding figure 4 above.**

**Reviewer 2**

Summary

Some models are more consistent with historical observations than others. In climate projection, it makes intuitive sense to give more weight to the models that are more consistent with observed climate shifts and less weight to models that are less consistent with observed climate shifts.

But how?

This paper reports on a method of assigning model weights that relies on two distinct distance measures: the distance of models from observations and the distance of models from other models. The method requires the specification of two parameters that determine how each of these distances are turned into model weights. The method for determining the parameter associated with inter-model distance is poorly explained(see specific comment 8 below). The method for determining the parameter associated with distance from observations is also poorly explained, but for many experiments, involves future-time-pseudo-observations from the future states that are the objective of the prediction (see comment 10 below). In other words, the tuning method appears to render the tests of the method to be of the "in-sample" variety. To weaken the degree of "in-sampleness" an additional test is performed using CMIP5 runs. However, since one expects many of the CMIP6 models to be closely related to the CMIP5 models,there are strong reasons to believe that this test is not truly "out-of-sample" either.

Even with the use of "future-time-pseudo-observations" in the tuning procedure, the improvements from this weighting scheme seem very modest in comparison with, for example, those obtained in Abramowitz and Bishop (2015, J. Climate) – (using a a method that solely required historical observations for the weights). The revised paper should include some attempt to compare/contrast/explain the Abramowitz and Bishop results.

A superficially appealing feature of the method is that it gives more weight to models that are both skillful and statistically independent of other models. However, this independence is just described in terms of inter-model distance and not in terms of the independence of the model error. Is there some unstated proof that increased inter-model distance equates to increased model error independence? (It seems easy to think of counter examples). As demonstrated in Bishop and Abramowitz (2013),it is the independence of the error of the individual models comprising an ensemble forecast (as measured by inter-model forecast error correlation) that increases the predictive power of the ensemble. The revised paper needs to address the issue of the relationship or lack of relationship between inter-model distance and model error independence.

After applying the method to the CMIP6 ensemble members, the authors find reduced warming relative to the simple sample means of CMIP6 ensembles for the high and low CO2 concentration scenarios considered. However, any confidence in this prediction must be strongly tempered by the "in sample" circular- nature of the testing and tuning procedures used by this method. My overall recommendation would be that the paper be returned to the authors to address the specific comments below and to include results from experiments in which only historical observations (or model-based-historical-pseudo-observations) were used to determine the weights. This constitutes major revision.

**We thank the reviewer for the critical assessment of our manuscript. The reviewer raises several important questions in the general comments above. Most of them we address in our answers to the specific comments as summarised below. In addition, we discuss the rationale behind our model independence metric in the following:**

- **Calculation of the independence shape parameter: see comment 8**

- **Calculation of the performance shape parameter: see comment 10**

- **Out of sample skill tests: see comment 10**

- **Skill improvement and comparison with Abramowitz and Bishop (2015): see comment 11; in addition we have added several references to the approach used therein in the revised manuscript.**

- **Model distance versus model error independence: see below**

**Model-model distance and model error correlation**

The weighting method we apply in our study separates between a model's performance and independence. For establishing either measure, different metrics have been used in the past (see line 145 in the revised manuscript). In the case of independence, one could, for example, argue that it should be based on our knowledge of a model's inner workings (such as shared components, parameterizations or heritage with other models). However, this information is not always easily accessible and is, in addition, hard to quantify. Therefore, we here use an output-based definition of independence: given a generalised distance metric (based on the climatology of two variables) we define independence as a model distance to all other models in the ensemble. This is equivalent to the distance of the models' errors:

$$e_i - e_j = (m_i - obs) - (m_j - obs) = m_i - m_j$$

where $e$ is the model error, $m$ is the model, and $obs$ the observation.

This approach has the advantage that it does not rely on observations, which are often geographically sparse and restricted in time. It, therefore, allows, in theory, establishing model independence based on hundreds of years of pre-industrial control runs or based on variables which do not have reliable global observations, such as precipitation.

Here we use surface air temperature and surface pressure as the basis for our estimate of independence. This follows the work of Merrifield et al. (2020), who show that using these two variables allow a clear separation of initial-condition members of the same model as well as closely related models on the one side and independent models on the other side (see, e.g., figure 5 in Merrifield et al., 2020). In addition, in our manuscript we show qualitative results of our independence classification as a model dependence tree in figure 5 and discuss several clusters where the "inner workings" are known (line 389-395 in the revised manuscript). As a further test we insert artificial new models into the ensemble (see figure 6 and related discussions). This allows us to investigate the change in independence weight based on the relation of the inserted model to the rest of the multi-model ensemble.

Bishop and Abramowitz (2013) follow a different approach that is based on the assumption that independent models have uncorrelated error time series. This approach can not directly be applied to our framework since we base our weighting on time-aggregated (mean, standard deviation, trend) spatially resolved fields. The main question the reviewer seems to pose, therefore is: Do the two approaches deliver fundamentally different results?

To test this we assume that the concept of error independence also holds for time-averaged spatial fields. We apply an independence weighting based on the spatial correlation of model errors and contrast the results with our original results (based on model distances). $S_{ij}$ in equation (1) then becomes the matrix of model error correlation distances:

$$S_{ij} = 1 - CORR_{spatial}(m_i - obs, m_j - obs)$$

Figure R1 below shows the models "family tree" equivalent to figure 5 in the manuscript based on these correlation distances. While the grouping of models is mostly the same as in figure 5,

there are also some obvious differences. The difference between the closest related models (e.g., UKESM1-0-LL and HadGEM3-GC1-LL) and the maximum distance between any two clusters of models is considerably larger. Several models have changed to a different cluster (e.g., NorESM2-MM or AWI-CM-1-1-MR). Without a detailed analysis, however, we can not make any clear statements on which clustering is "more correct".

[Figure]

**Figure R1: Similar to figure 5 in the revised manuscript but based on error correlation distances instead of model-model distances. Note that for this case we do not use any area weighting.**

Based on the general similarity of the two trees, we do not expect the change in the independence metric to have a major influence on the results. In a second step we, therefore, look at the weighted distributions based on independence weights using these error correlation distances. The results are presented in figure R2 below. Compared to figure 8 in the revised manuscript there are only minimal differences. This at least shows that there are no strong disagreements between the approaches. One reason for the similarity is certainly also the fact that the weighting is dominated to a large degree by the performance weighting and, in particular, by the low weights of some of the strong warming models.

In summary we, therefore, argue that either approach might be appropriate to use, and the main conclusions in our manuscript are the same for an independence matric based on correlation. For simplicity we, therefore, prefer to continue using our original metric basing independence directly on model-model distances which does not require observations and thus eliminates one potential source of uncertainty. We have, however, added section S4 to the supplement discussing or method to estimate model independence in the context of other approaches.

[Figure]

**Figure R2: Similar to figure 8 in the revised manuscript but with the independence weighting based on error correlation distances instead of model-model distances. Note that for this case we do not use any area weighting in the independence weighting calculation.**

Specific comments

1. Line 16. Consider explaining what TCR is in the abstract to appeal to a broader audience.

**Indeed our study aims at a quite general audience and therefore focuses mainly on projections of future global warming which are widely known. In the revised manuscript we no longer mention TCR in the abstract.**

2. Line 31. Do you mean model uncertainty, unknown model climate error, unknown model-climate-sensitivity-to-CO2 error or model climate differences? We know what the model is, and we can determine its climate past, present and future by running it. We can also determine the differences between the climates of different models. Given the limitations of the spatio-temporal distribution of observations, the uncertain thing is the actual climate both past, present, and future, is it not?

**Model uncertainty here refers to the error of both present and future climate. In particular to its bias, since for climate projections we are concerned with correctly estimating distributions of trajectories, rather than individual trajectories like for weather and climate prediction.**

**"Model uncertainty" has become a standard piece of terminology in this subfield, following its popularization by Hawkins and Sutton (2009). It is also mentioned as "structural" uncertainty or error, referring to the structure of the model (which is assumed to be different between different climate models, hence the "model" label). We have updated the paragraph in question to make that more clear (lines 30-34 in the revised manuscript).**

3. Line 35. Lorenz, the father of chaos theory, argued that while the accuracy of weather forecasts was limited to a few weeks the climate of a system was not sensitive to specified initial conditions and could be known provided the forcing on the system was known. I guess "climate" in the sense of Lorenz refers to the statistical description of the attractor of the chaotic system. When you refer to "internal variability" do you just mean slow modes of the model's chaotic attractor that might possibly be confused with a change in the mean of the model's attractor if the ensemble size was too small?

**"Internal variability" indeed refers to initial condition sensitivity; the terminology has become standard in the climate literature following papers like Hawkins and Sutton (2009) or Deser et**

al. (2012). Here "climate" refers to the statistical description of the attractor of the system which these models attempt to represent - including the atmosphere but also the ocean, ice, and land surface. Particularly for the ocean, coupled models and the real earth's coupled system show variations on timescales of, at the very least, multiple years (e.g., due to ENSO) that depend on the initial conditions. Recently some efforts have sought to identify predictability on the order of decades, though if this exists it is assumed (here and generally) to be small.

Because GCMs are expensive to run and have unknown but expected long timescales before ensemble variance that properly samples the climatology is achieved, CMIP models are not at the point where many of them have enough ensemble members to adequately sample the attractor (in contrast to weather prediction, where that is currently achievable and in fact often achieved). With a small number of ensemble members and long timescales, internal variability is convolved with forced responses. These can be isolated with "large ensembles" (of several tens of simulations differing only by initial conditions) but the CMIP ensemble includes many models which are expected to differ in their bias, some of which also include multiple realizations from the same model, which are expected to differ among each other only in terms of their "internal variability" or due to sampling. We have added some discussion to the paragraph in question (lines 34-40 in the revised manuscript).

Deser, C., Phillips, A. S., Bourdette, V., & Teng, H. (2012). Uncertainty in climate change projections: the role of internal variability. Climate Dynamics, 38(3–4), 527–546. https://doi.org/10.1007/s00382-010-0977-x

4. Line 102: I'm guessing you are referring to Section 3.2 of Brunner et al., 2019. Is that correct? If so, please state this in the text. Your wording suggested that you had estimated an observation error variance. However, on reading Section 3.2 of Brunner et al., 2019, I'm now guessing that you are referring to how your derived weights change depending on which subset of all observations you use. Are you suggesting that the reason for your weights changing is because the observations have different errors? Can you rule out the possibility that your weighting scheme isn't just over-fitting each individual observational data set? In any case, the revised paper needs to clarify whether in fact you are referring to the size of the change in weights associated with using differing observational data sets. Also, the observed values are known. They are not uncertain. The errors of the observed values are unknown. It is the observational error that is uncertain.

Thank you for pointing this out, the wording was unclear in the original manuscript. Indeed, it has been pointed out in the literature that using different observational datasets can lead to diverging results in some cases (e.g., Gleckler et al. 2008, Lorenz et al. 2018, Brunner et al. 2019) due to differences in the datasets. We referred to these differences in the observational datasets as observational uncertainty but no longer do so in the revised manuscript.

What we are concerned with here is bias in the observational datasets, which are a central challenge in climate science. In the presence of such biases, it is not unexpected that the results of the weighting change based on the datasets used. To get a reference that is as robust as possible, we are using a combination of two observational datasets (ERA5 and MERRA2) to calculate the model-observation distances and further the performance weights. The datasets are combined by taking the center of the observational spread at each grid cell (following Brunner et al. 2019 who also discuss other approaches; see their section 3.2 as well as section S2 and figure S3 in their supplement). We have clarified that and added additional information to the section in question in the revised manuscript (lines 103-110).

Gleckler, P. J., Taylor, K. E., & Doutriaux, C. (2008). Performance metrics for climate models. Journal of Geophysical Research Atmospheres, 113(6), 1–20. https://doi.org/10.1029/2007JD008972

5. Line 145. "We want to..." If there was a hypothetical user of the climate projection that only cared about temperature trend and not about year-to-year variability, might you not be doing them a

disservice by down-weighting members that have an excellent temperature trend but poor inter-annual variability? Consider changing to "We choose to..."

**The reviewer is correct in pointing out that the selection of diagnostics for establishing the models performance weights should depend on the target in question. In our study we look at temperature change in two time periods as a target, which is closely related to the temperature trend. Therefore, the temperature trend is indeed a powerful diagnostic.**

**However, it also is strongly influenced by internal variability (i.e., it differs quite strongly between initial-condition members of the same model) which is not desirable for a good diagnostic as we argue in line 171 of the revised manuscript: "*Ideally, a performance weight is reflective of underlying model properties and does not depend on which ensemble member is chosen to represent that model (i.e., on internal variability). tasTREND does not fulfil this requirement: the spread within one model is the same order of magnitude as the spread among different models.*"**

**We therefore use "*a balanced combination of climate system features (i.e., diagnostics) relevant for the target to inform the weighting to minimise the risk for skill decreases. This guards against the possibility of a model "accidentally" fitting observations for a single diagnostic while being far away from them in several others (and hence possibly not providing a skilful projection of the target variable).*" (line 454 of the revised manuscript)**

**In this sense we argue that even if a user is only interested in a model simulating future temperature trend correctly, it might still be important to also include other diagnostics. This can help to avoid weighting a model highly because it "accidentally" matches the observations in a given historical period due to, e.g., internal variability.**

6. Line 147-149. Equations should be added to precisely describe these observation derived quantities – perhaps in an appendix or supplementary material.

**We have now added a mathematical description of the diagnostic calculation to the supplement (section S2) and reference it in the revised manuscript in line 160.**

7. Line 170. You must state what was used as a proxy for a perfect model. I would think that the derived sigma_D must be related to the ensemble variance of the model states around the time averaged state. That quantity will depend on the model will it not? Please clarify.

**We have adjusted our description of the shape parameter calculation in the revised manuscript in order to make this more clear. In the revised manuscript we now refer to the iterative test used to the performance shape parameter as parameter calibration (lines 182-191). In addition we have added additional information including a schematic of the calibration test to the supplement (section S3).**

8. Line 183. I looked at Section 2.3 of Brunner et al., 2019 for an explanation but Brunner et al. (2019) just directs the reader to Lorenz et al., 2018. Your work needs to be reproducible. When referring to another paper for a key explanation, you must give very specific information about where in the paper the explanation resides (e.g. a section number) to ensure reproducibility. You have not done this.

**The reviewer rightfully points out that we should have been more clear in referencing this important information. The calculation of the independence shape parameter and reasoning behind it is described in detail in the supplement of Brunner et al. (2019; section S3.1), which we now explicitly mention. In addition we now provide a summary as well as a discussion of the chosen value in the context of our study in the supplement of the revised manuscript (see line 200 and supplement section S5 in the revised manuscript).**

9. Line 191. The method used to evaluate performance given here seems almost identical to that given in Abramowitz and Bishop (2015) but no reference is given to this paper or others that may have used this approach before. Such literature is relevant and should be cited.

**Thank you for pointing this out. We have added several references to the relevant literature which used similar approaches before (see line 206 in the revised manuscript).**

10. Line 200-205. Here, we learn that sigma_D weights are determined in part from information from a place that is inaccessible in reality: the future. Only model futures are accessible. By line 205 we learn that the model future states (rather than observations) are, in fact, an integral part of choosing the weights. This is a significant departure from many other observation-based methods for improving ensemble forecasts and projections. The use of future time observations in the training causes all of the associated tests to be "in-sample" tests – dramatically reducing their trustworthiness. Since the CMIP5 models belong to the same general class of human produced climate simulators they can barely be considered "out-of-sample". Please comment on the limitations of this approach. In addition, you have not clarified how the method of tuning for future states interacts with the method to determine sigma_D referred to on line 170 (see previous comment).

**The reviewer rightfully points out that there is some influence from the future model states included in the weights via the performance parameter calibration. However, there also seems to be some misunderstanding regarding our approach. We adapted the sections in question to make it more clear in the revised manuscript.**

**The model performance weights are proportional to each model's generalised distance (a combination of 5 diagnostics) to the observations ($D_i$) as given in the numerator of equation (1). The proportionality constant is the performance shape parameter sigma_D, which translates these distances into the weights. It is indeed established using the target period, i.e., the future model states. The weighting for the ensemble is then calibrated as a whole using this single parameter, and it is not the case that the weight of each model is calibrated individually through its historical simulation.**

**Crucially, this means that the weighting is still dominated by the comparison of models to the observations only. Consider, for example, a case where the diagnostics are really poorly chosen: this could be because they are dominated by (random) internal variability or because they do not have any physical relationship to the target. The weighting then would not have any skill, regardless of the sigma_D parameter.**

**As, for example, Sanderson et al. (2017) state, selecting sigma_D only based on historical information might lead to overconfident results as a more skillful representation of the base state does not necessarily translate to a more skillful representation of the future. Selecting sigma_D only based on historical information would a priori assume that the chosen metric is relevant for the projection. One way of approaching the problem might be to apply the method on the historical and then test the result in a perfect model test, potentially adjusting the method in an iterative approach to maximise skill.**

**In our weighting approach we already include such a perfect model test in the calculation of the weights in order to avoid overconfident results. To avoid confusion between the setting of the parameter and the subsequent testing of method skill we have changed the terminology in our manuscript and refer to the former as *parameter calibration* to separate it from the later perfect model tests which are used to calculate the skill of the weighting. In addition we have added a section in the supplement detailing and visualising this parameter calibration (section S3 and figure S1 in the revised manuscript).**

**Finally, addressing the question of the relationship between the calibration of the performance shape parameter and the subsequent testing of the skill of the method, we would argue that the circularity is quite limited. There are several reasons for this:**

- As we point out above, the weighting is, to a large degree, based on the model's distance to historical observations, with future observations only influencing them via sigma_D, which is a single value constant across all models, over time, and all metrics.

- The parameter calibration does not aim at maximising (mean) skill, but rather ensures that the results are not overconfident. Take the example of poorly chosen diagnostics again: in such a case, any separation into better or worse models would be overconfident as it would be based on pure chance. During the parameter calibration this would become obvious and sigma_D would be relaxed to a large value (in order to avoid this overconfidence) leading to an approximation of equal weighting. Subsequently testing the skill of the method can still be insightful to estimate the actual increase in skill (or the lack thereof - in the case of badly chosen diagnostics).

- We use two different model pools to draw the perfect models from in our investigation of the method's skill. The first one is based on CMIP6 data, and one could therefore argue that it has a stronger potential circularity as the same models have been used to calibrate sigma_D. However, this test is mainly used to investigate the relative differences between different combinations of diagnostics and to select the best performing one (see figure 1 and related discussion). Since any remaining circularity is the same for all cases shown in figure 1, a comparison between them should still be valid. We have adapted the abstract as well as section 3.1 to make that more clear.

- For the second test, we use CMIP5 models, which have not been used in the parameter calibration, as perfect models. Here, another potential issue arises: several CMIP6 models are related to CMIP5 models and are therefore not independent. However, about eight years of additional model development lie between the two generations. In addition, it has been noted that several CMIP6 models have a much higher climate sensitivity and are, hence, quite different from their predecessors (at least in their response to anthropogenic forcing, which dominates the future period used for the perfect model test).

To further increase the independence between the CMIP5 and CMIP6 ensembles, we now exclude directly-related models from the perfect model test in the revised manuscript. So, for example, when weighting based on the CMIP5 model HadGEM2-ES we exclude the CMIP6 models HadGEM3-GC31-LL and UKESM1-0-LL from the evaluation. A list of CMIP6 models excluded for each CMIP5 model can be found in table S5 in the supplement and we have added some discussion about this topic in section S3 of the supplement.

11. Line 266-280. Here we learn that the method is very prone to creating decreased skill relative to the multi-model unweighted mean. This negative result is in contrast to the positive results found in Abramowitz and Bishop (2015) using the method of Bishop and Abramowitz (2013).

Thank you for pointing this out, this was not expressed clearly in the original manuscript. In fact, the method produces a median skill increase of about 12-22% when using CMIP5 models as pseudo observations (see figure 3a in the revised manuscript). Nonetheless, it is correct that there can be a decrease in skill from the unweighted to the weighted multi-model ensemble based on our skill metric when using some CMIP5 models as pseudo-observations. However, these instances are limited to only a few (about 15% across SSPs and target periods) cases. We have revised the paragraph in question to make this more clear (line 314-322 in the revised manuscript).

We note that the change in skill also depends on the skill metric used and the target it is applied to. Here, our target is 20-year mean, global mean temperature change from 1995-2014 to two future periods (2041-60 and 2081-00). As a skill metric, we use the continuous ranked probability skill score (CRPSS), a measure for ensemble forecast quality. Note that this does not only evaluate the distance between the (un-) weighted mean and the reference but also considers the full distribution.

[revised manuscript text omitted]